# GradSkip: Communication-Accelerated Local Gradient Methods with Better Computational Complexity

**Artavazd Maranjyan**[*]
*King Abdullah University of Science and Technology (KAUST)*

*arto.maranjyan@gmail.com*

**Mher Safaryan**
*King Abdullah University of Science and Technology (KAUST)*

*mher.safaryan@gmail.com*

**Peter Richtárik**
*King Abdullah University of Science and Technology (KAUST)*

*peter.richtarik@kaust.edu.sa*

**Reviewed on OpenReview:** *https://openreview.net/forum?id=6R3fRqFfhn*

## Abstract

We study a class of distributed optimization algorithms that aim to alleviate high communication costs by allowing clients to perform multiple local gradient-type training steps before communication. In a recent breakthrough, Mishchenko et al. (2022) proved that local training, when properly executed, leads to provable communication acceleration, and this holds in the strongly convex regime without relying on any data similarity assumptions. However, their ProxSkip method requires all clients to take the same number of local training steps in each communication round. We propose a redesign of the ProxSkip method, allowing clients with "less important" data to get away with fewer local training steps without impacting the overall communication complexity of the method. In particular, we prove that our modified method, GradSkip, converges linearly under the same assumptions and has the same accelerated communication complexity, while the number of local gradient steps can be reduced relative to a local condition number. We further generalize our method by extending the randomness of probabilistic alternations to arbitrary unbiased compression operators and by considering a generic proximable regularizer. This generalization, which we call GradSkip+, recovers several related methods in the literature as special cases. Finally, we present an empirical study on carefully designed toy problems that confirm our theoretical claims.

## 1 Introduction

*Federated Learning (FL)* is an emerging distributed machine learning paradigm where diverse data holders or clients (e.g., smartwatches, mobile devices, laptops, hospitals) collectively aim to train a single machine learning model without revealing local data to each other or the orchestrating central server (McMahan et al., 2017; Kairouz et al, 2019; Wang, 2021). Training such models amounts to solving federated optimization problems of the form

$$\min_{x \in \mathbb{R}^d} \left\{ f(x) := \frac{1}{n} \sum_{i=1}^{n} f_i(x) \right\}, \tag{1}$$

where $d$ is the (typically large) number of parameters of the model $x \in \mathbb{R}^d$ we aim to train, and $n$ is the (potentially large) total number of devices in the federated environment. We denote by $f_i(x)$ the loss or risk associated with the data $\mathcal{D}_i$ stored on client $i \in [n] := \{1, 2, \dots, n\}$. Formally, our goal is to minimize the overall loss/risk denoted by $f(x)$.

---

[*]Work done during an internship at KAUST, while being a researcher at YerevaNN and a student at Yerevan State University.

Due to their efficiency, *gradient-type methods* with its numerous extensions (Duchi et al., 2011; Zeiler, 2012; Ghadimi & Lan, 2013; Kingma & Ba, 2015; Schmidt et al., 2017; Qian et al., 2019; Gorbunov et al., 2020a) is by far the most dominant method for solving (1) in practice.

The simplest implementation of gradient descent in a federated setup requires all workers $i \in [n]$ in each time step $t \geq 0$ to

(i) compute the local gradient $\nabla f_i(x_t)$ at the current global model $x_t$,

(ii) update the current global model $x_t$ using this gradient with step size $\gamma > 0$

$$\hat{x}_{i,t+1} = x_t - \gamma \nabla f_i(x_t), \tag{2}$$

(iii) average the updated local models $\hat{x}_{i,t+1}$ to get the new global model

$$x_{t+1} = \frac{1}{n} \sum_{i=1}^{n} \hat{x}_{i,t+1}. \tag{3}$$

Challenges defining FL as a unique distributed training setup, necessitating training algorithm adjustments, include *high communication costs*, *heterogeneous data distribution*, and *system heterogeneity* across clients. Next, we discuss these challenges and potential algorithmic solutions.

## 1.1 Communication Costs

In federated optimization, communication costs often become a primary bottleneck due to slow and unreliable wireless links between clients and the central server (McMahan et al., 2017). Eliminating the communication step (3) entirely would cause clients to train solely on local data, leading to a poor model because of the limited local data.

A simple trick to reduce communication costs is to perform the costly synchronization step (3) infrequently, allowing multiple local gradient steps (2) in each communication round (Mangasarian, 1995). This trick appears in the celebrated FedAvg algorithm of McMahan et al. (2016; 2017) and its further variations (Haddadpour & Mahdavi, 2019; Li et al., 2019a; Khaled et al., 2019a;b; Karimireddy et al., 2020; Horváth et al., 2022) under the name of *local gradient methods*. However, until very recently, theoretical guarantees on the convergence rates of local gradient methods were worse than the rate of classical gradient descent, which synchronizes after every gradient step.

In a recent line of works (Mishchenko et al., 2022; Malinovsky et al., 2022; Condat & Richtárik, 2022; Sadiev et al., 2022), initiated by Mishchenko et al. (2022), a novel local gradient method, called ProxSkip, was proposed which performs *a random number* of local gradient steps before each communication (alternation between local training and synchronization is probabilistic) and guarantees strong communication acceleration properties. First, they reformulate the problem (1) into an equivalent regularized consensus problem of the form

$$\min_{x_1,\ldots,x_n \in \mathbb{R}^d} \left\{ \frac{1}{n} \sum_{i=1}^{n} f_i(x_i) + \psi(x_1,\ldots,x_n) \right\},$$

$$\psi(x_1,\ldots,x_n) := \begin{cases} 0, & \text{if } x_1 = \cdots = x_n, \\ +\infty, & \text{otherwise}, \end{cases} \tag{4}$$

where communication between the clients and averaging local models $x_1,\ldots,x_n$ is encoded as taking the proximal step with respect to $\psi$, i.e.,

$$\mathrm{prox}_{\psi}([x_1 \cdots x_n]^{\top}) = [\bar{x} \cdots \bar{x}]^{\top}, \quad \text{where} \quad \bar{x} := \frac{1}{n} \sum_{i=1}^{n} x_i.$$

With this reformulation, ProxSkip by Mishchenko et al. (2022) performs the proximal (equivalently averaging) step with small probability $p = 1/\sqrt{\kappa}$, where $\kappa$ is the condition number of the problem. Then method's key result

for smooth, strongly convex setups is $\mathcal{O}(\kappa \log {}^1/_\epsilon)$ iteration complexity with $\mathcal{O}\left(\sqrt{\kappa} \log {}^1/_\epsilon\right)$ communication rounds to achieve $\epsilon > 0$ accuracy. Follow-up works extend the method to variance-reduced gradient methods (Malinovsky et al., 2022), randomized application of proximal operator (Condat & Richtárik, 2022), and accelerated primal-dual algorithms (Sadiev et al., 2022). Our work was inspired by the development of this new generation of local gradient methods, also known as Local Training (LT) methods, which we detail shortly.

An orthogonal approach uses communication compression strategies on the transferred information. Informally, instead of communicating full precision models infrequently, we might communicate a compressed version of the local model in each iteration via an application of lossy compression operators. Such strategies include sparsification (Alistarh et al., 2018; Wang et al., 2018; Mishchenko et al., 2020), quantization (Alistarh et al., 2017; Sun et al., 2019; Wang et al., 2022), sketching (Hanzely et al., 2018; Safaryan et al., 2021) and low-rank approximation (Vogels et al., 2019).

Our work contributes to the first approach to handling high communication costs that is less understood in theory and, at the same time, immensely popular in the practice of FL.

## 1.2 Statistical Heterogeneity

Due to the decentralized training data, distributions of local datasets can vary from client to client. This heterogeneity in data distributions poses an additional challenge since allowing multiple local steps would make the local models deviate from each other, an issue widely known as *client drift*. On the other hand, if training datasets are identical across the clients (commonly referred to as a homogeneous setup), the mentioned drifting issue disappears, and the training can be done without any communication whatsoever. Interpolating between these extremes, under some data similarity conditions (which are typically expressed as gradient similarity conditions), multiple local gradient steps should be useful. In fact, initial theoretical guarantees of local gradient methods utilize such assumptions (Haddadpour & Mahdavi, 2019; Yu et al., 2019; Li et al., 2019b; 2020).

In the fully heterogeneous setup, client drift reduction techniques were designed and analyzed to mitigate the adverse effect of local model deviations (Karimireddy et al., 2020; Gorbunov et al., 2021). A very close analogy is variance reduction techniques called error feedback mechanisms for the compression noise added to lessen the number of bits required to transfer (Condat et al., 2022).

## 1.3 System Heterogeneity

Lastly, system heterogeneity refers to the diversity of clients in terms of their computation capabilities or the amount of resources they are willing to use during the training. In a typical FL setup, all participating clients must perform the same amount of local gradient steps before each communication. Consequently, a highly heterogeneous cluster of devices results in significant and unexpected delays due to slow clients or stragglers.

One approach addressing system heterogeneity or dealing with slow clients is client selection strategies (Wang & Joshi, 2019; Reisizadeh et al., 2020; Luo et al., 2021). Basically, client sampling can be organized so that slow clients do not delay global synchronization, and clients with similar computational capabilities are sampled in each communication round.

Unlike the above strategy, we suggest clients take local steps based on their resources. We consider the full participation setup where clients decide how much local computation to perform before communication. Informally, slow clients do less local work than fast clients, and during the synchronization of locally trained models, the slowdown caused by the stragglers will be minimized see section 5.2.

## 1.4 Local Training (LT) vs Accelerated Gradient Descent (AGD)

Nesterov's AGD method Nesterov (2004) matches the communication complexity of our GradSkip algorithm. Its distributed implementation takes one local step per round, suggesting LT methods might lag behind AGD. In contrast, almost all methods in production are based on local training, as evidenced by FL frameworks like He et al. (2020); Ro et al. (2021); Beutel et al. (2022).

The preference for LT over AGD among practitioners stems from LT's advantages, especially in generalization and communication complexity. Both areas are closely tied with local training, becoming prominent in current research. LT's ability to enhance generalization remains under exploration in FL. Current studies link this improvement to personalization, meta-learning Hanzely et al. (2020); Hanzely & Richtárik (2021), and representation learning Collins et al. (2022). Practically, LT effectively tackles nonconvex challenges, while AGD faces difficulty approximating stationary points of smooth nonconvex functions. Additionally, AGD is more sensitive to the knowledge of the condition number than LT methods, which are versatile and work across a wide range of numbers of local steps.

In statistically heterogeneous cases, AGD often underperforms. Our experiments prove this by showing that when device condition numbers vary, AGD converges slower than GradSkip. Though our work does not primarily aim to directly compare AGD and LT, such a comparative study, to our knowledge, remains a gap in current research and could offer valuable insights.

## 2 Summary of Contributions

Our key contributions are summarized below.

### 2.1 GradSkip: Efficient Gradient Skipping Algorithm

We propose GradSkip (Algorithm 1), a new local gradient-type method for distributed optimization that reduces both communication and computation. Our method extends the recently developed ProxSkip (Mishchenko et al., 2022), which first demonstrated communication acceleration via multiple local steps without data similarity assumptions. GradSkip not only inherits this accelerated communication complexity but also introduces a key improvement: *it allows clients to terminate their local gradient computations independently*, significantly improving computational efficiency.

The key technical novelty of the proposed algorithm is the construction of auxiliary shifts $\hat{h}_{i,t}$ to handle gradient skipping for each client $i \in [n]$. GradSkip also maintains shifts $h_{i,t}$ initially introduced in ProxSkip to handle communication skipping across the clients. We prove that GradSkip converges linearly in strongly convex and smooth setup, has the same $\mathcal{O}(\sqrt{\kappa_{\max}} \log 1/\epsilon)$ accelerated communication complexity as ProxSkip, and requires clients to compute (in expectation) at most $\min\left\{\kappa_i, \sqrt{\kappa_{\max}}\right\}$ local gradients in each communication round (see Theorem 3.6), where $\kappa_i$ is the condition number for client $i \in [n]$ and $\kappa_{\max} = \max_i \kappa_i$. Thus, for GradSkip, clients with well-conditioned problems $\kappa_i < \sqrt{\kappa_{\max}}$ perform much less local work to achieve the same convergence rate of ProxSkip, which assumes $\sqrt{\kappa_{\max}}$ local steps on average for all clients.

### 2.2 GradSkip+: General GradSkip Method

Next, we generalize the construction and the analysis of GradSkip by extending it in two directions: handling optimization problems with arbitrary proximable regularizer and incorporating general randomization procedures using unbiased compression operators with custom variance bounds. This leads to our second method, GradSkip+ (see Algorithm 2), which recovers several methods in the literature as a special case, including the standard proximal gradient descent (ProxGD), ProxSkip (Mishchenko et al., 2022), RandProx-FB (Condat & Richtárik, 2022) and GradSkip.

### 2.3 VR-GradSkip+: Reducing the Variance of Stochastic Gradient Skipping

Finally, we propose and analyze variance-reduced extension (see Algorithm 3 in the Appendix) in the case when mini-batch stochastic gradients are implemented instead of full-batch gradients for local computations. Our VR-GradSkip+ method can be viewed as a successful combination of ProxSkip-VR method of Malinovsky et al. (2022) and GradSkip providing computational efficiency through processing smaller batch of samples and probabilistically skipping stochastic gradient computations. We deferred the presentation of the part of our contribution in the appendix due to space limitations.

# 3 GradSkip

In this section, we present our first algorithm, GradSkip, and discuss its benefits in detail. Later, we will generalize it, unifying several other methods as special cases. Recall that our target is to address three challenges in FL mentioned in the introductory part, which are

    (i) reduction in communication cost via infrequent synchronization of local models,

    (ii) statistical or data heterogeneity, and

    (iii) reduction in computational cost via limiting local gradient calls based on the local subproblem.

We now describe all the steps of the algorithm and how it handles these three challenges.

## 3.1 Algorithm Structure

For the sake of presentation, we describe the progress of the algorithm using two variables $x_{i,t}, \hat{x}_{i,t}$ for the local models and two variables $h_{i,t}, \hat{h}_{i,t}$ for the local gradient shifts. Essentially, we want to maintain two variables for the local models since clients get synchronized infrequently. The shifts $h_{i,t}$ are designed to reduce the client drift caused by the statistical heterogeneity. Finally, we introduce auxiliary shifts $\hat{h}_{i,t}$ to take care of the different number of local steps. The GradSkip method is formally presented in Algorithm 1.

---

**Algorithm 1** GradSkip

1: **Input:** stepsize $\gamma > 0$, synchronization probability $p$, probabilities $q_i > 0$ controlling local steps, initial local iterates $x_{1,0} = \cdots = x_{n,0} \in \mathbb{R}^d$, initial shifts $h_{1,0}, \ldots, h_{n,0} \in \mathbb{R}^d$, total number of iterations $T \geq 1$
2: **for** $t = 0, 1, \ldots, T - 1$ **do**
3:    **server:** Flip a coin $\theta_t \in \{0, 1\}$ with $\mathrm{Prob}(\theta_t = 1) = p$       ⋄ Decide when to skip communication
4:    **for all devices $i \in [n]$ in parallel do**
5:       Flip a coin $\eta_{i,t} \in \{0, 1\}$ with $\mathrm{Prob}(\eta_{i,t} = 1) = q_i$       ⋄ Decide when to skip gradient steps
                                                                             (see Lemma 3.1)
6:       $\hat{h}_{i,t+1} = \eta_{i,t} h_{i,t} + (1 - \eta_{i,t}) \nabla f_i(x_{i,t})$       ⋄ Update the local auxiliary shifts $\hat{h}_{i,t}$
7:       $\hat{x}_{i,t+1} = x_{i,t} - \gamma(\nabla f_i(x_{i,t}) - \hat{h}_{i,t+1})$       ⋄ Update the local auxiliary iterate $\hat{x}_{i,t}$
                                                                                  via shifted gradient step
8:       **if** $\theta_t = 1$ **then**
9:          $x_{i,t+1} = \frac{1}{n} \sum_{j=1}^{n} \left( \hat{x}_{j,t+1} - \frac{\gamma}{p} \hat{h}_{j,t+1} \right)$       ⋄ Average shifted iterates, but only very rarely!
10:       **else**
11:          $x_{i,t+1} = \hat{x}_{i,t+1}$       ⋄ Skip communication!
12:       **end if**
13:       $h_{i,t+1} = \hat{h}_{i,t+1} + \frac{p}{\gamma}(x_{i,t+1} - \hat{x}_{i,t+1})$       ⋄ Update the local shifts $h_{i,t}$
14:    **end for**
15: **end for**

---

As an initialization step, we choose a probability $p > 0$ to control communication rounds, probabilities $q_i > 0$ for each client $i \in [n]$ to control local gradient steps, and initial control variates (or shifts) $h_{i,0} \in \mathbb{R}^d$ to control client drift. Besides, we fix the stepsize $\gamma > 0$ and assume that all clients commence with the same local model, namely $x_{1,0} = \cdots = x_{n,0} \in \mathbb{R}^d$. Then, each iteration of the method comprises two stages, the local stage and the communication stage, operating probabilistically. Specifically, the probabilistic nature of these stages is the following. The local stage requires computation only with some predefined probability; otherwise, the stage is void. Similarly, the communication stage requires synchronization between all clients only with probability $p$; otherwise, the stage is void. In the local stage (lines 5–7), all clients $i \in [n]$ in parallel update their local variables $(\hat{x}_{i,t+1}, \hat{h}_{i,t+1})$ using values $(x_{i,t}, h_{i,t})$ from previous iterate either by computing the local gradient $\nabla f_i(x_{i,t})$ or by just copying the previous values. Afterward, in the communication stage (lines 8–13), all clients in parallel update their local variables $(x_{i,t+1}, h_{i,t+1})$ from $(\hat{x}_{i,t+1}, \hat{h}_{i,t+1})$ by either averaging across the clients or copying previous values.

## 3.2 Reduced Local Computation

Clearly, communication costs are reduced as the averaging step occurs only when $\theta_t = 1$ with probability $p$ of our choice. However, it is not directly apparent how the computational costs are reduced during the local stage. Indeed, both options $\eta_{i,t} = 1$ and $\eta_{i,t} = 0$ involve the expression $\nabla f_i(x_{i,t})$ as if local gradients need to be evaluated in every iteration. As we show in the following lemma, this is not the case.

**Lemma 3.1** (Fake local steps; Proof in Appendix C.1). *Suppose that Algorithm 1 does not communicate for $\tau \geq 1$ consecutive iterates, i.e., $\theta_t = \theta_{t+1} = \cdots = \theta_{t+\tau-1} = 0$ for some fixed $t \geq 0$. Besides, let for some client $i \in [n]$ we have $\eta_{i,t} = 0$. Then, regardless of the coin tosses $\{\eta_{i,t+j}\}_{j=1}^{\tau}$, client $i$ does fake local steps without any gradient computation in $\tau$ iterates. Formally, for all $j = 1, 2, \ldots, \tau + 1$, we have*

$$\hat{x}_{i,t+j} = x_{i,t+j} = x_{i,t},$$
$$\hat{h}_{i,t+j} = h_{i,t+j} = h_{i,t} = \nabla f_i(x_{i,t}).$$

Let us reformulate the above lemma. During the local stage of GradSkip, when clients do not communicate with the server, $i^{th}$ client terminates its local gradient steps once the local coin tosses $\eta_{i,t} = 0$. Thus, smaller probability $q_i$ implies sooner coin toss $\eta_{i,t} = 0$ in expectation, hence, less amount of local computation for client $i$. Therefore, we can relax the computational requirements of clients by adjusting these probabilities $q_i$ and controlling the amount of local gradient computations.

Next, let us find out how the expected number of local gradient steps depends on probabilities $p$ and $q_i$. Let $\Theta$ and $H_i$ be random variables representing the number of coin tosses (Bernoulli trials) until the first occurrence of $\theta_t = 1$ and $\eta_{i,t} = 0$ respectively. Equivalently, $\Theta \sim \text{Geo}(p)$ is a geometric random variable with parameter $p$, and $H_i \sim \text{Geo}(1 - q_i)$ are geometric random variables with parameter $1 - q_i$ for $i \in [n]$. Notice that, within one communication round, $i^{th}$ client performs $\min\{\Theta, H_i\}$ number of local gradient computations, which is again a geometric random variable with parameter $1 - (1 - (1 - q_i))(1 - p) = 1 - q_i(1 - p)$. Therefore, as formalized in the next lemma, the expected number of local gradient steps is $\mathbb{E}[\min\{\Theta, H_i\}] = 1/(1 - q_i(1 - p))$.

**Lemma 3.2** (Expected number of local steps; Proof in Appendix C.2). *The expected number of local gradient computations in each communication round of GradSkip is $1/(1 - q_i(1 - p))$ for all clients $i \in [n]$.*

Notice that, in the special case of $q_i = 1$ for all $i \in [n]$, GradSkip recovers Scaffnew method of Mishchenko et al. (2022). However, as we will show, we can choose probabilities $q_i$ smaller, reducing computational complexity and obtaining the same convergence rate as Scaffnew.

*Remark* 3.3 (System heterogeneity). From this discussion, we conclude that GradSkip can also address system or device heterogeneity. In particular, probabilities $\{q_i\}_{i=1}^{n}$ can be assigned to clients in accordance with their local computational resources; slow clients with scarce compute power should get small $q_i$, while faster clients with rich resources should get bigger $q_i \leq 1$, see section 5.2.

## 3.3 Convergence Theory

Now that we explained the structure and computational benefits of the algorithm, let us proceed to the theoretical guarantees. We consider the same strongly convex and smooth setup as considered by Mishchenko et al. (2022) for the distributed case.

**Assumption 3.4.** All functions $f_i(x)$ are strongly convex with parameter $\mu > 0$ and have Lipschitz continuous gradients with Lipschitz constants $L_i > 0$, i.e., for all $i \in [n]$ and any $x, y \in \mathbb{R}^d$ we have

$$\frac{\mu}{2}\|x - y\|^2 \leq D_{f_i}(x, y) \leq \frac{L_i}{2}\|x - y\|^2,$$

where $D_{f_i}(x, y) := f_i(x) - f_i(y) - \langle \nabla f_i(y), x - y \rangle$ is the Bregman divergence associated with $f_i$.

We present Lyapunov-type analysis to prove the convergence, which is a very common approach for iterative algorithms. Consider the Lyapunov function

$$\Psi_t := \sum_{i=1}^{n} \|x_{i,t} - x_\star\|^2 + \frac{\gamma^2}{p^2} \sum_{i=1}^{n} \|h_{i,t} - h_{i,\star}\|^2, \tag{5}$$

where $\gamma > 0$ is the stepsize, $x_\star$ is the (necessary) unique minimizer of $f(x)$ and $h_{i,*} = \nabla f_i(x_*)$ is the optimal gradient shift. As we show next, $\Psi_t$ decreases at a linear rate.

**Theorem 3.5** (Proof in Appendix C.3). *Let Assumption 3.4 hold. If the stepsize satisfies*

$$\gamma \leq \min_i \left\{ \frac{1}{L_i} \frac{p^2}{1 - q_i(1 - p^2)} \right\}$$

*and probabilities are chosen so that $0 < p$, $q_i \leq 1$, then the iterates of* GradSkip *(Algorithm 1) satisfy*

$$\mathbb{E}[\Psi_t] \leq (1 - \rho)^t \Psi_0, \tag{6}$$

*for all $t \geq 1$ with $\rho := \min\left\{ \gamma\mu, 1 - q_{max}(1 - p^2) \right\} > 0$.*

Let us comment on this result.

- The first and immediate observation from the above result is that, with a proper stepsize choice, GradSkip converges linearly for any choice of probabilities $p$ and $q_i$ from $(0, 1]$.

- Furthermore, by choosing all probabilities $q_i = 1$ we get the same rate of Scaffnew with $\rho = \min\{\gamma\mu, p^2\}$ (see Theorem 3.6 in (Mishchenko et al., 2022)). If we further choose the largest admissible stepsize $\gamma = 1/L_{\max}$ and the optimal synchronization probability $p = 1/\sqrt{\kappa_{\max}}$, we get $\mathcal{O}(\kappa_{\max} \log 1/\epsilon)$ iteration complexity, $\mathcal{O}(\sqrt{\kappa_{\max}} \log 1/\epsilon)$ accelerated communication complexity with $1/p = \sqrt{\kappa_{\max}}$ expected number of local steps in each communication round. Here, we used notation $\kappa_{\max} = \max_i \kappa_i$ where $\kappa_i = L_i/\mu$ is the condition number for client $i \in [n]$.

- Finally, exploiting smaller probabilities $q_i$, we can optimize computational complexity subject to the same communication complexity as Scaffnew. To do that, note that the largest possible stepsize that Theorem 3.5 allows is $\gamma = 1/L_{\max}$ as

$$\min_i \left\{ \frac{1}{L_i} \frac{p^2}{1 - q_i(1 - p^2)} \right\} \leq \min_i \frac{1}{L_i} = \frac{1}{L_{\max}}.$$

  Hence, taking into account $\rho \leq \gamma\mu$, the best iteration complexity from the rate (6) is $\mathcal{O}(\kappa_{\max} \log 1/\epsilon)$, which can be obtained by choosing the probabilities appropriately as formalized in the following result.

**Theorem 3.6** (Optimal parameter choices; Proof in Appendix C.5). *Let Assumption 3.4 hold and choose probabilities*

$$q_i = \frac{1 - \frac{1}{\kappa_i}}{1 - \frac{1}{\kappa_{\max}}} \leq 1 \quad and \quad p = \frac{1}{\sqrt{\kappa_{\max}}}.$$

*Then, with the largest admissible stepsize $\gamma = 1/L_{\max}$,* GradSkip *enjoys the following properties:*

*(i) $\mathcal{O}\left(\kappa_{\max} \log 1/\varepsilon\right)$ iteration complexity,*

*(ii) $\mathcal{O}\left(\sqrt{\kappa_{\max}} \log 1/\varepsilon\right)$ communication complexity,*

*(iii) for each client $i \in [n]$, the expected number of local gradient computations per communication round is*

$$\frac{1}{1 - q_i(1 - p)} = \frac{\kappa_i(1 + \sqrt{\kappa_{\max}})}{\kappa_i + \sqrt{\kappa_{\max}}} \leq \min\left\{ \kappa_i, \sqrt{\kappa_{\max}} \right\}. \tag{7}$$

This result clearly quantifies the benefits of using smaller probabilities $q_i$. In particular, if the condition number $\kappa_i$ of client $i$ is smaller than $\sqrt{\kappa_{\max}}$, then within each communication round, it does only $\kappa_i$ number of local gradient steps. However, for a client having the maximal condition number (namely, clients $\arg\max_i\{\kappa_i\}$), the number of local gradient steps is $\sqrt{\kappa_{\max}}$, which is the same for Scaffnew. From this, we conclude that, in terms of computational complexity, GradSkip is always better and can be $\mathcal{O}(n)$ times better than Scaffnew (Mishchenko et al., 2022).

# 4 GradSkip+

We extend GradSkip in two directions, leading to our general GradSkip+ method. The first extension concerns the optimization problem formulation. As discussed, the distributed problem (1) with consensus constraints can be reformulated as a regularized problem (4) in a lifted space, where the local variables $x_1, \ldots, x_n \in \mathbb{R}^d$ are stacked into a single vector in $\mathbb{R}^{nd}$. Following Mishchenko et al. (2022), we consider the lifted problem

$$\min_{x \in \mathbb{R}^d} f(x) + \psi(x), \tag{8}$$

where $f(x)$ is a smooth, strongly convex loss, and $\psi(x)$ is a closed, proper, convex regularizer (see (4)). We require that the proximal operator of $\psi$ is a single-valued function that can be computed.

The second extension in GradSkip+ is the generalization of the randomization procedure of probabilistic alternations in GradSkip by allowing arbitrary unbiased compression operators with certain bounds on the variance. Let us formally define the class of compressors we will be working with.

**Definition 4.1** (Unbiased Compressors). For any positive semidefinite matrix $\mathbf{\Omega} \succeq 0$, denote by $\mathbb{B}^d(\mathbf{\Omega})$ the class of (possibly randomized) unbiased compression operators $\mathcal{C} \colon \mathbb{R}^d \to \mathbb{R}^d$ such that for all $x \in \mathbb{R}^d$ we have

$$\mathbb{E}\left[\mathcal{C}(x)\right] = x,$$
$$\mathbb{E}\left[\left\|(\mathbf{I} + \mathbf{\Omega})^{-1}\mathcal{C}(x)\right\|^2\right] \leq \|x\|^2_{(\mathbf{I}+\mathbf{\Omega})^{-1}}.$$

The class $\mathbb{B}^d(\mathbf{\Omega})$ is a generalization of commonly used class $\mathbb{B}^d(\omega)$ of unbiased compressors with variance bound $\mathbb{E}\left[\|\mathcal{C}(x)\|^2\right] \leq (1+\omega)\|x\|^2$ for some scalar $\omega \geq 0$. Indeed, when the matrix $\mathbf{\Omega} = \omega\mathbf{I}$, then $\mathbb{B}^d(\omega\mathbf{I})$ coincides with $\mathbb{B}^d(\omega)$. Furthermore, the following inclusion holds:

**Lemma 4.2** (Proof in Appendix D.1). $\mathbb{B}^d(\mathbf{\Omega}) \subseteq \mathbb{B}^d\left(\frac{(1+\lambda_{\max}(\mathbf{\Omega}))^2}{(1+\lambda_{\min}(\mathbf{\Omega}))} - 1\right)$.

The purpose of this new variance bound with matrix parameter $\mathbf{\Omega}$ is to introduce non-uniformity on the level of compression across different directions. For example, in the reformulation (4) each client controls $1/n$ portion of the directions and the level of compression. For example, consider compression operator $\mathcal{C} \colon \mathbb{R}^d \to \mathbb{R}^d$ defined as

$$\mathcal{C}(x)_j = \begin{cases} x_j/p_j, & \text{with probability } p_j, \\ 0, & \text{with probability } 1 - p_j, \end{cases} \tag{9}$$

for all coordinates $j \in [d]$ and for any $x \in \mathbb{R}^d$, where $p_j \in (0, 1]$ are given probabilities. Then, it is easy to check that $\mathcal{C} \in \mathbb{B}^d(\mathbf{\Omega})$ with diagonal matrix $\mathbf{\Omega} = \mathbf{Diag}(1/p_j - 1)$ having diagonal entries $1/p_j - 1 \geq 0$.

With finer control over the compression operator, we can use the granular smoothness information of the loss function $f$ via smoothness matrices (Qu & Richtárik, 2016b;a).

**Definition 4.3** (Matrix Smoothness). A differentiable function $f : \mathbb{R}^d \to \mathbb{R}$ is called $\mathbf{L}$-smooth with some symmetric and positive definite matrix $\mathbf{L} \succ 0$ if

$$D_f(x, y) \leq \frac{1}{2}\|x - y\|^2_{\mathbf{L}}, \quad \forall x, y \in \mathbb{R}^d, \tag{10}$$

where $\|x\|_{\mathbf{L}} := \sqrt{x^\top \mathbf{L} x}$ denotes the $\mathbf{L}$-norm of $x$.

The standard $L$-smoothness condition with scalar $L > 0$ is obtained as a special case of (10) for matrices of the form $\mathbf{L} = L\mathbf{I}$, where $\mathbf{I}$ is the identity matrix. The notion of matrix smoothness provides more information about the function than mere scalar smoothness. In particular, if $f$ is $\mathbf{L}$-smooth, then it is also $\lambda_{\max}(\mathbf{L})$-smooth due to the relation $\mathbf{L} \preceq \lambda_{\max}(\mathbf{L})\mathbf{I}$. Smoothness matrices have been used in the literature of randomized coordinate descent (Richtárik & Takáč, 2016; Hanzely & Richtárik, 2019b;a) and distributed optimization (Safaryan et al., 2021; Wang et al., 2022).

---

**Algorithm 2** GradSkip+

---

1: **Parameters:** stepsize $\gamma > 0$, compressors $\mathcal{C}_\omega \in \mathbb{B}^d(\omega)$ and $\mathcal{C}_\Omega \in \mathbb{B}^d(\Omega)$.
2: **Input:** initial iterate $x_0 \in \mathbb{R}^d$, initial control variate $h_0 \in \mathbb{R}^d$, number of iterations $T \geq 1$.
3: **for** $t = 0, 1, \ldots, T-1$ **do**
4:    $\hat{h}_{t+1} = \nabla f(x_t) - (\mathbf{I} + \mathbf{\Omega})^{-1} \mathcal{C}_\Omega (\nabla f(x_t) - h_t)$       $\diamond$ Update the shift $\hat{h}_t$ via shifted compression
5:    $\hat{x}_{t+1} = x_t - \gamma(\nabla f(x_t) - \hat{h}_{t+1})$       $\diamond$ Update the iterate $\hat{x}_t$ via a shifted gradient step
6:    $\hat{g}_t = \frac{1}{\gamma(1+\omega)} \mathcal{C}_\omega \left( \hat{x}_{t+1} - \mathrm{prox}_{\gamma(1+\omega)\psi} \left( \hat{x}_{t+1} - \gamma(1+\omega)\hat{h}_{t+1} \right) \right)$       $\diamond$ Estimate the proximal gradient
7:    $x_{t+1} = \hat{x}_{t+1} - \gamma\hat{g}_t$       $\diamond$ Update the main iterate $x_t$
8:    $h_{t+1} = \hat{h}_{t+1} + \frac{1}{\gamma(1+\omega)}(x_{t+1} - \hat{x}_{t+1})$       $\diamond$ Update the main shift $h_t$
9: **end for**

---

## 4.1 Algorithm Description

Similar to GradSkip, we maintain two variables $x_t$, $\hat{x}_t$ for the model, and two variables $h_t$, $\hat{h}_t$ for the gradient shifts in GradSkip+. Initial values $x_0 \in \mathbb{R}^d$ and $h_0 \in \mathbb{R}^d$ can be chosen arbitrarily. In each iteration, GradSkip+ first updates the auxiliary shift $\hat{h}_{t+1}$ using the previous shift $h_t$ and gradient $\nabla f(x_t)$ (line 4). This shift $\hat{h}_{t+1}$ is then used to update the auxiliary iterate $x_t$ via shifted gradient step (line 5). Then we estimate the proximal gradient $\hat{g}_t$ (line 6) in order to update the main iterate $x_{t+1}$ (line 7). Lastly, we complete the iteration by updating the main shift $h_t$ (line 8). See Algorithm 2 for the formal steps.

## 4.2 Special Cases

GradSkip+ recovers several existing methods as special cases, including ProxGD, ProxSkip, and RandProx-FB (Condat & Richtárik, 2022).

- **ProxGD.** When $\mathcal{C}_\omega$ is the identity compressor (i.e., $\omega = 0$), then Algorithm 2 reduces to the ProxGD algorithm as

$$x_{t+1} = \mathrm{prox}_{\gamma\psi}(\hat{x}_{t+1} - \gamma\hat{h}_{t+1}) = \mathrm{prox}_{\gamma\psi}(x_t - \gamma\nabla f(x_t))$$

  for any choice of $\mathcal{C}_\Omega$.

- **ProxSkip.** Let $\mathcal{C}_\Omega$ be the identity compressor (i.e., $\mathbf{\Omega} = \mathbf{I}$) and $\mathcal{C}_\omega$ be the Bernoulli compressor $\mathcal{C}_p$ with parameter $p \in (0, 1]$ (note that here $\omega = 1/p - 1$). In this case, $\hat{h}_{t+1} \equiv h_t$ and

$$x_{t+1} = \begin{cases} \mathrm{prox}_{\frac{\gamma}{p}\psi} \left( \hat{x}_{t+1} - \frac{\gamma}{p}h_t \right), & \text{with probability } p, \\ \hat{x}_{t+1}, & \text{with probability } 1 - p. \end{cases}$$

  Thus, we recover the ProxSkip algorithm.

- **RandProx-FB.** Let $\mathcal{C}_\Omega$ be the identity compressor and $\mathcal{C}_\Omega = \mathcal{R} \in \mathbb{B}^d(\omega)$. Then, after the following change of notation:

$$h_t = -u_t, \quad \hat{g}_t = \frac{d_t}{1 + \omega_2},$$

  the method is equivalent to RandProx-FB (Condat & Richtárik, 2022), which is a generalization of ProxSkip when additional smoothness information for the regularizer $\psi$ is known[1].

- **GradSkip.** Finally, we can specialize GradSkip+ to recover GradSkip. Consider the lifted space $\mathbb{R}^{nd}$ where $x \in \mathbb{R}^{nd}$ represents the concatenations of models $x_1, \ldots, x_n \in \mathbb{R}^d$ from all clients. The central example of an unbiased compression operator for that would be the probabilistic switching mechanism used in GradSkip, which is sometimes referred to as Bernoulli compressor: for any given $p \in [0, 1]$, the compressor outputs

$$\mathcal{C}_p^{nd}(x) = \begin{cases} \frac{x}{p}, & \text{with probability } p, \\ 0, & \text{with probability } 1 - p, \end{cases}$$

---

[1] We do not consider smooth regularizers as our primary example of regularizer is the non-smooth consensus constraint (4).

for any input vector $x \in \mathbb{R}^{nd}$. GradSkip employs one Bernoulli compressor $\mathcal{C}_p^{nd}$ with parameter $p \in (0, 1]$ controlling communication rounds, and one Bernoulli compressor $\mathcal{C}_{q_i}^d$ with parameter $q_i \in (0, 1]$ for each client to control local gradient steps. Therefore, choosing $\mathcal{C}_\omega = \mathcal{C}_p^{nd}$ and $\mathcal{C}_\mathbf{\Omega} = \mathcal{C}_{q_1}^d \times \cdots \times \mathcal{C}_{q_n}^d$ in the lifted space $\mathbb{R}^{nd}$, GradSkip+ reduces to GradSkip.

### 4.3 Convergence Theory

We now present the convergence theory for GradSkip+, for which we replace the scalar smoothness Assumption 3.4 by matrix smoothness.

**Assumption 4.4** (Convexity and smoothness)**.** We assume that the loss function $f$ is $\mu$-strongly convex with positive $\mu > 0$ and $\mathbf{L}$-smooth with positive definite matrix $\mathbf{L} \succ 0$.

Similar to (5), we analyze GradSkip+ using the Lyapunov function

$$\Psi_t := \|x_t - x_\star\|^2 + \gamma^2(1 + \omega)^2 \|h_t - \nabla f(x_*)\|^2.$$

The next theorem shows the linear convergence result.

**Theorem 4.5** (Proof in Appendix D.2)**.** *Let Assumption 4.4 hold, $\mathcal{C}_\omega \in \mathbb{B}^d(\omega)$ and $\mathcal{C}_\mathbf{\Omega} \in \mathbb{B}^d(\mathbf{\Omega})$ be the compression operators, and*

$$\widetilde{\mathbf{\Omega}} := \mathbf{I} + \omega(\omega + 2)\mathbf{\Omega}(\mathbf{I} + \mathbf{\Omega})^{-1}.$$

*Then, if the stepsize $\gamma \leq \lambda_{\max}^{-1}(\mathbf{L}\widetilde{\mathbf{\Omega}})$, the iterates of GradSkip+ (Algorithm 2) satisfy*

$$\mathbb{E}\left[\Psi_t\right] \leq \left(1 - \min\left\{\gamma\mu, \delta\right\}\right)^t \Psi_0, \tag{11}$$

*where*

$$\delta = 1 - \frac{1}{1 + \lambda_{\min}(\mathbf{\Omega})}\left(1 - \frac{1}{(1 + \omega)^2}\right) \in [0, 1].$$

First, if we choose $\mathcal{C}_\mathbf{\Omega}$ to be the identity compression (i.e., $\mathbf{\Omega} = \mathbf{0}$), then GradSkip+ reduces to RandProx-FB, and we recover asymptotically the same rate with linear factor $(1 - \min\{\gamma\mu, 1/(1+\omega)^2\})$ (see Theorem 3 of Condat & Richtárik (2022)). If we further choose $\mathcal{C}_\omega$ to be the Bernoulli compression with parameter $p \in (0, 1]$, then $\omega = 1/p - 1$ and we get the rate of ProxSkip.

To recover the rate in (6) for GradSkip, consider the lifted space $\mathbb{R}^{nd}$ and the reformulated problem (4) with objective $f(x) = \frac{1}{n}\sum_{i=1}^n f_i(x_i)$, where $x_i \in \mathbb{R}^d$ and $x = (x_1, \ldots, x_n) \in \mathbb{R}^{nd}$. Since each $f_i$ is $\mu$-strongly convex, the function $f$ is also $\mu$-strongly convex. Regarding the smoothness condition, we have $L_i\mathbf{I} \in \mathbb{R}^{d\times d}$ smoothness matrices (e.g., scalar $L_i$-smoothness) for each $f_i$, which implies that the overall loss function $f$ has $\mathbf{L} = \mathbf{Diag}(L_1\mathbf{I}, \ldots, L_n\mathbf{I}) \in \mathbb{R}^{nd\times nd}$ as a smoothness matrix.

Furthermore, choosing Bernoulli compression operators $\mathcal{C}_\omega = \mathcal{C}_p^{nd}$ and $\mathcal{C}_\mathbf{\Omega} = \mathcal{C}_{q_1}^d \times \cdots \times \mathcal{C}_{q_n}^d$ in the lifted space $\mathbb{R}^{nd}$, we get

$$\omega = \frac{1}{p} - 1 \quad \text{and} \quad \mathbf{\Omega} = \mathbf{Diag}\left(\frac{1}{q_i} - 1\right).$$

It remains to plug all these expressions into Theorem 4.5 and recover Theorem 3.6. Indeed, $\lambda_{\min}(\mathbf{\Omega}) = 1/q_{\max} - 1$ and, hence, $\delta = 1 - q_{\max}\left(1 - p^2\right)$.

Finally, Theorem 4.5 gives the same stepsize bound

$$\lambda_{\max}^{-1}(\mathbf{L}\widetilde{\mathbf{\Omega}}) = \min_i\left\{L_i\left(1 + (1 - q_i)\left(\frac{1}{p^2} - 1\right)\right)\right\}^{-1} = \min_i\left\{\frac{1}{L_i}\frac{p^2}{1 - q_i\left(1 - p^2\right)}\right\}.$$

## 5 Experiments

To test the performance of GradSkip and illustrate theoretical results, we use the classical logistic regression problem.

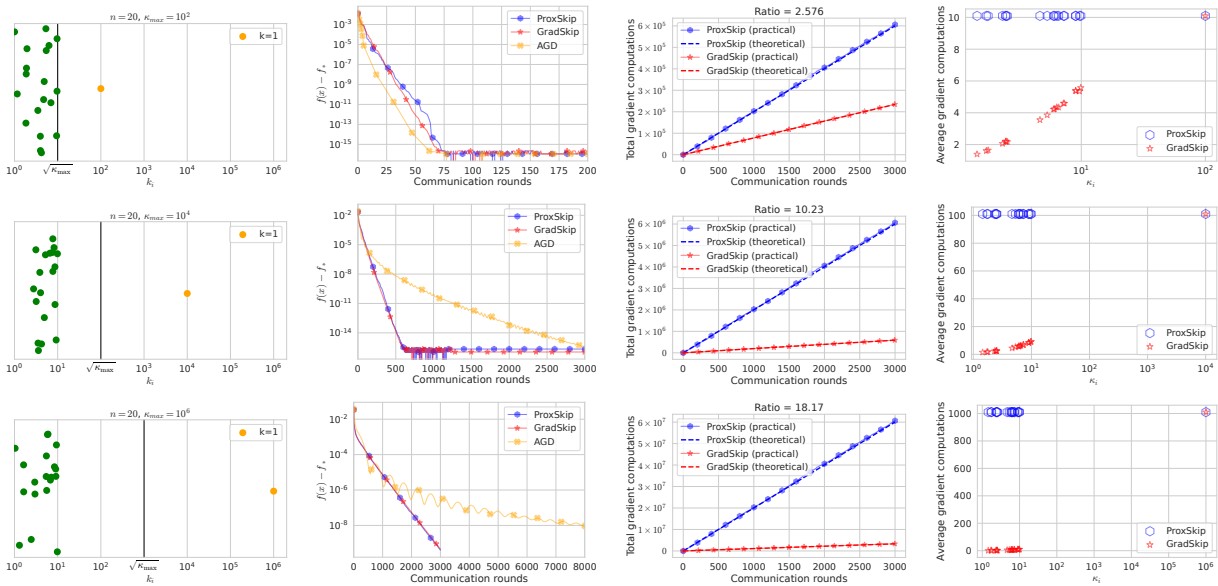

Figure 1: The first column displays the condition numbers for devices. The second column presents convergence per communication round. The third column contrasts theoretical and practical gradient computation counts. The final column reveals the average gradient computations for devices with condition number $\kappa_i$. Notably, in GradSkip, the device with $\kappa_i = \kappa_{max}$ performs gradient computations at a rate comparable to all devices in ProxSkip.

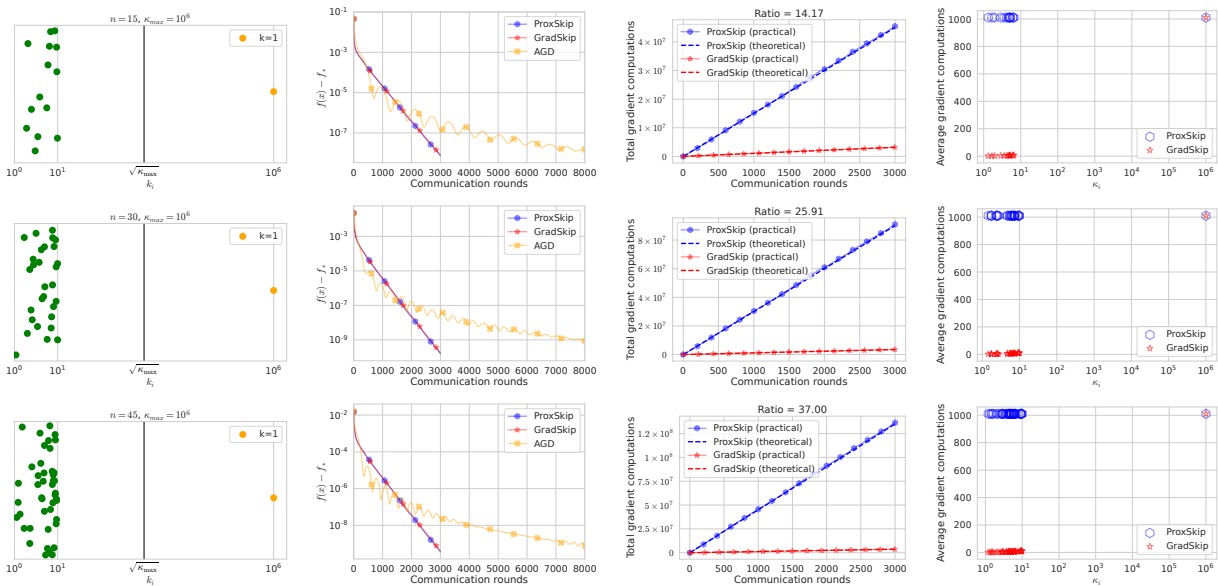

Figure 2: The columns in this figure represent the same as those in Figure 1.

The loss function for this model has the following form:

$$f(x) = \frac{1}{n} \sum_{i=1}^{n} \frac{1}{m} \sum_{j=1}^{m} \log\left(1 + \exp\left(-b_{ij} a_{ij}^{\top} x\right)\right) + \frac{\lambda}{2} \|x\|^2,$$

where $n$ is the number of clients, $m$ is the number of data points per worker, $a_{ij} \in \mathbb{R}^d$ and $b_{ij} \in \{-1, +1\}$ are the data samples, and $\lambda$ is the regularization parameter.

Experiments were conducted on artificially generated data and on the *"australian"* dataset from LibSVM library (Chang & Lin, 2011) (see Section 5.1). All algorithms were run using their theoretically optimal hyperparameters (stepsize, probabilities). We compare GradSkip with ProxSkip and AGD, which have SOTA accelerated communication complexity. Comparisons between VR-GradSkip+ and ProxSkip-VR were omitted, as their computational complexity difference is similar to that of GradSkip and ProxSkip.

For GradSkip, the expected local gradient computations per communication round are at most $\sum_{i=1}^{n} \min\left(\kappa_i, \sqrt{\kappa_{\max}}\right)$ (see (7)), while for ProxSkip, it is $n\sqrt{\kappa_{\max}}$. Therefore, the gradient computation ratio of ProxSkip over GradSkip depends on the number of devices with $\kappa_i \geq \sqrt{\kappa_{\max}}$. With $k \leq n$ such devices, this ratio for ProxSkip over GradSkip converges to $n/k \geq 1$ as $\kappa_{\max} \to \infty$.

In our experiments, only one device has an ill-conditioned local problem ($k = 1$). To showcase this convergence, we generate data to control the smoothness constants and set the regularization parameter $\lambda = 10^{-1} = \mu$. We run GradSkip and ProxSkip algorithms for 3000 communication rounds. Figure 1 features $n = 20$ devices, one with a large $L_i = L_{max}$, and others with $L_i \sim \text{Uniform}(0.1, 1)$. The second column illustrates similar convergence for GradSkip and ProxSkip. As we increment $L_{\max}$ row by row, the ratio converges to $n = 20$, while AGD's performance drops with increasing data heterogeneity. Figure 2 illustrates the growing ratio with more clients $n$, assigning one device $L_i = L_{max} = 10^5$ and others $L_i \sim \text{Uniform}(0.1, 1)$, showing the increase in $n$ row by row.

## 5.1 Experiment on the *"australian"* Dataset

In line with our experiments on synthetic data (section 5), we conduct a parallel experiment using the *"australian"* dataset from the LibSVM library (Chang & Lin, 2011). This involves applying the GradSkip and ProxSkip algorithms to the logistic regression problem, characterized by the same loss function used previously:

$$f(x) = \frac{1}{n} \sum_{i=1}^{n} \frac{1}{m} \sum_{j=1}^{m} \log\left(1 + \exp\left(-b_{ij} a_{ij}^{\top} x\right)\right) + \frac{\lambda}{2} |x|^2.$$

We set the regularization parameter $\lambda = 10^{-4} L_{\max}$. We split the dataset equally into $n = 20$ devices. In this case we get $k = 8$ devices with ill-conditioned local problems, so the gradient computation ratio of ProxSkip over GradSkip should be close to $n/k = 2.5$. It can be seen in Figure 3.

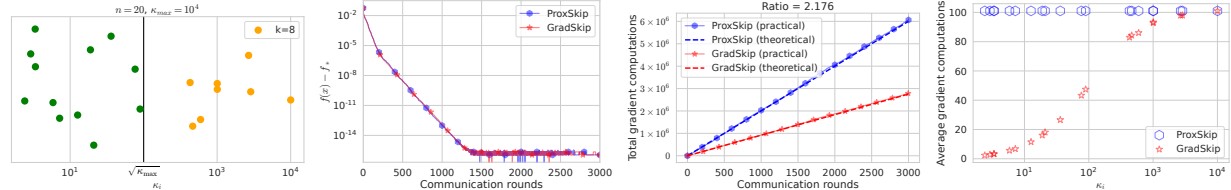

Figure 3: The plots have the same meaning as in Figure 1.

## 5.2 System Heterogeneity Case

Let $T_i$ represent the time required for client $i$ to complete one local step. We consider $T_i$ to be a random variable with the structure $T_i = \tau_i + \eta_i$. Here, $\tau_i$ is a scalar representing the minimum time to finish one local step on machine $i$, and $\eta_i$ is a random jitter (time delay) assumed to have an exponential distribution with scale parameter $\beta_i$. Practically, the distribution of $T_i$ can be estimated.

Our objective is to determine values for $q_i$ that minimize the wall training time (excluding communication time) in GradSkip. The average expected time for local training before communication on client $i$ is:

$$\frac{\mathbb{E}\left[T_i\right]}{1 - q_i(1 - p)},$$

given that, on average, device $i$ performs

$$\frac{1}{1 - q_i(1 - p)}$$

local steps (see Lemma 3.2). To reduce waiting time, we initiate by setting $q_i = 1$ for the fastest clients. For other clients, we set $q_i$ to make the average local training time before communication match with the fastest device. This condition can be mathematically expressed as:

$$\frac{\mathbb{E}\left[T_i\right]}{1 - q_i(1 - p)} = \frac{\mathbb{E}\left[T_{min}\right]}{p},$$

yielding the value of $q_i$ as

$$q_i = \max\left\{\frac{1 - p\frac{\mathbb{E}[T_i]}{\mathbb{E}[T_{min}]}}{1 - p}, 0\right\}.$$

To assess the effectiveness of our $q_i$ selection strategy in GradSkip compared to ProxSkip, we ran experiments with two types of delay distributions for $\tau_i$: uniform and exponential. In the first case, $\tau_i \sim \mathrm{Uniform}(0, 1)$, and in the second, $\tau_i \sim \mathrm{Exponential}(1)$. To introduce variability in communication delays, we also added noise $\eta_i \sim \mathrm{Exponential}(\beta_i)$ with $\beta_i \sim \mathrm{Uniform}(0, 1)$.

The results, shown in Figure 4, demonstrate that GradSkip, with adaptively chosen $q_i$, achieves better performance than ProxSkip with a fixed $q$ under both delay models.

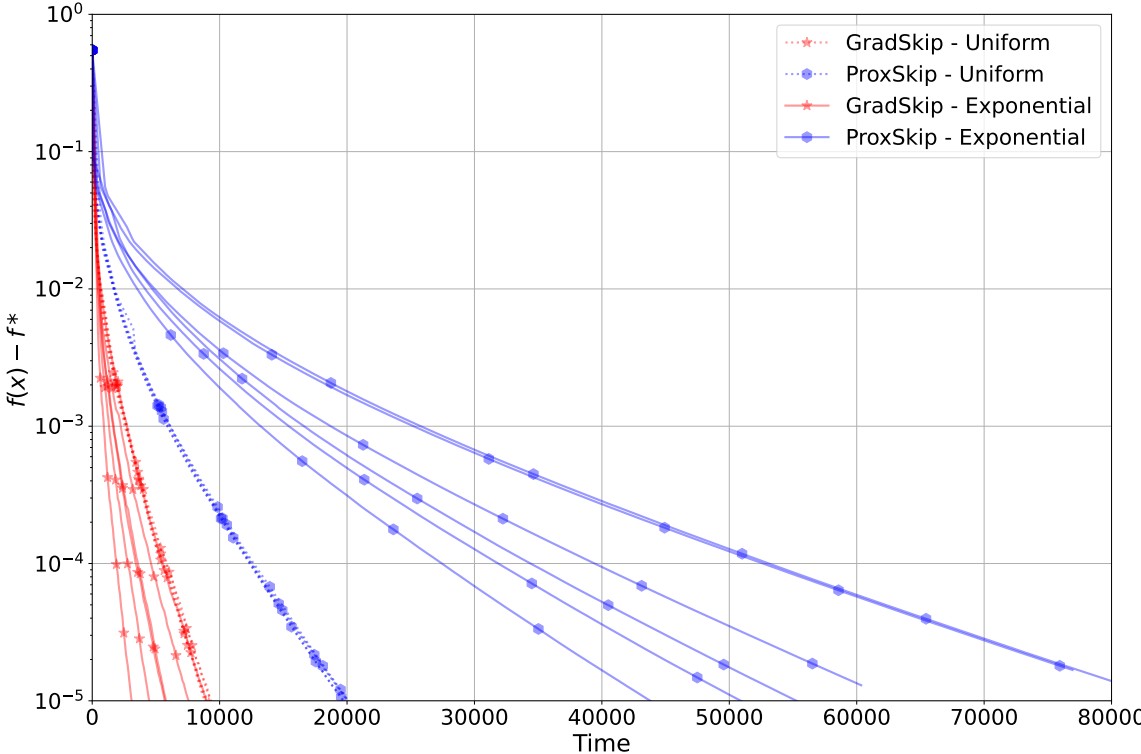

Figure 4: We used the *"w6a"* dataset from the LibSVM library (Chang & Lin, 2011), which has $d = 300$ features. The number of clients is 153.[2]

---

[2]There is no particular reason for this choice, other than that 153 is a "nice" number: $153 = 1! + 2! + 3! + 4! + 5! = 1^3 + 5^3 + 3^3$.

**Acknowledgments**

The research reported in this publication was supported by funding from King Abdullah University of Science and Technology (KAUST): i) KAUST Baseline Research Scheme, ii) Center of Excellence for Generative AI, under award number 5940, iii) SDAIA-KAUST Center of Excellence in Artificial Intelligence and Data Science.

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

# A   Limitations and Future Work

In this part, we outline some limitations and future research directions related to our work.

- Similar to the previous works Malinovsky et al. (2022); Mishchenko et al. (2022) on local gradient methods with communication acceleration, our theory does not cover non-strongly convex or non-convex objective functions. So far, the communication acceleration property of local steps has been proven only for a strongly convex setup.

- Another key component for designing efficient distributed and federated learning algorithms is partial device participation. This extension seems rather tricky, and we leave this as a future work. A recent work by Grudzień et al. (2023) considers client sampling.

- Finally, one can combine the local gradient methods with communication compression techniques to achieve even better communication complexity. Moreover, our proposed gradient skipping approach can be decoupled to address computational complexity, too.

# B   Extension to Stochastic Gradients with Variance Reduction: VR-GradSkip+

Recently developed ProxSkip-VR method (Malinovsky et al., 2022) reduces computational complexity by allowing computationally cheaper stochastic gradient estimators instead of full batch gradients. This approach of reducing computational complexity is blind to statistical heterogeneity and is entirely orthogonal to our approach of reducing computational complexity in GradSkip. It is natural to ask the following question.

> *Is it possible to combine these two methods (ProxSkip-VR and GradSkip) to achieve even better computational complexity?*

We give an affirmative answer to the question by developing our most general VR-GradSkip+ method.

## B.1   Algorithm Description

We get VR-GradSkip+ method from GradSkip+ by replacing the gradient $\nabla f(x_t)$ by an unbiased estimator

$$g_t = \text{StochasticGradient}(x_t, f),$$

see Algorithm 3.

Our next assumption, initially introduced by Gorbunov et al. (2020a), postulates several parametric inequalities characterizing the behavior and, ultimately, the quality of a gradient estimator. Similar assumptions appeared later in (Gorbunov et al., 2020b; 2021).

**Assumption B.1.** Let $\{x_t\}$ be the iterates produced by VR-GradSkip+. We first assume unbiasedness of the stochastic gradients $g_t$ for all iterations $t \geq 0$, i.e.,

$$\mathbb{E}\left[g_t \mid x_t\right] = \nabla f(x_t).$$

Next, we assume that for some non-negative constants $A, B, C, \tilde{A}, \tilde{B}, \tilde{C}$, with $\tilde{B} < 1$, and non-negative sequence $\{\sigma_t\}_{t \geq 0}$ the following inequalities hold for all $t \geq 0$:

$$
\begin{aligned}
\mathbb{E}\left[\|g_t - \nabla f(x_\star)\|^2_{\mathbf{L}^{-1}} \mid x_t\right] &\leq 2A D_f(x_t, x_\star) + B\sigma_t + C, \\
\mathbb{E}\left[\sigma_{t+1} \mid x_t\right] &\leq 2\tilde{A} D_f(x_t, x_\star) + \tilde{B}\sigma_t + \tilde{C}.
\end{aligned}
$$

Assumption B.1 covers a very large collection of gradient estimators, including an infinite variety of sub-sampling/minibatch estimators, gradient sparsification and quantization estimators, and their combinations; see (Gorbunov et al., 2020a) for examples. VR estimators are characterized by $C = \tilde{C} = 0$; most non-VR estimators by $\tilde{A} = \tilde{B} = \tilde{C} = B = 0$ and $C > 0$ (Gower et al., 2019).

---

**Algorithm 3** VR-GradSkip+

---

1: **Parameters:** stepsize $\gamma > 0$, compressors $\mathcal{C}_\omega \in \mathbb{B}^d(\omega)$ and $\mathcal{C}_\mathbf{\Omega} \in \mathbb{B}^d(\mathbf{\Omega})$.
2: **Input:** initial iterate $x_0 \in \mathbb{R}^d$, initial control variate $h_0 \in \mathbb{R}^d$, number of iterations $T \geq 1$.
3: **for** $t = 0, 1, \ldots, T-1$ **do**
4:     $g_t = \text{StochasticGradient}(x_t, f)$        $\diamond$ Construct an unbiased estimator of $\nabla f(x_t)$
5:     $\hat{h}_{t+1} = g_t - (\mathbf{I} + \mathbf{\Omega})^{-1}\mathcal{C}_\mathbf{\Omega}\,(g_t - h_t)$        $\diamond$ Update the shift $\hat{h}_t$ via shifted compression
6:     $\hat{x}_{t+1} = x_t - \gamma(g_t - \hat{h}_{t+1})$        $\diamond$ Update the iterate $\hat{x}_t$ via shifted stochastic gradient step
7:     $\hat{g}_t = \frac{1}{\gamma(1+\omega)}\mathcal{C}_\omega\left(\hat{x}_{t+1} - \text{prox}_{\gamma(1+\omega)\psi}\left(\hat{x}_{t+1} - \gamma(1+\omega)\hat{h}_{t+1}\right)\right)$    $\diamond$ Estimate the proximal gradient
8:     $x_{t+1} = \hat{x}_{t+1} - \gamma\hat{g}_t$        $\diamond$ Update the main iterate $x_t$
9:     $h_{t+1} = \hat{h}_{t+1} + \frac{1}{\gamma(1+\omega)}(x_{t+1} - \hat{x}_{t+1})$        $\diamond$ Update the main shift $h_t$
10: **end for**

---

## B.2 Convergence Theory

Consider the Lyapunov function:

$$\Psi_t := \|x_t - x_\star\|^2 + \gamma^2(1+\omega)^2\|h_t - h_\star\|^2 + \gamma^2 W\sigma_t,$$

where $h_* = \nabla f(x_*)$.

**Theorem B.2** (Proof in Appendix B.4). *Let Assumption 4.4 hold, and let $g_t$ be a gradient estimator satisfying Assumption B.1. Let $\mathcal{C}_\omega \in \mathbb{B}^d(\omega)$ and $\mathcal{C}_\mathbf{\Omega} \in \mathbb{B}^d(\mathbf{\Omega})$ be the compression operators. If $B > 0$, choose any*

$$W > \frac{\lambda_{\max}(\mathbf{L}\widetilde{\mathbf{\Omega}})B}{1 - \tilde{B}} \quad and \quad \beta = 1 - \tilde{B} - \frac{\lambda_{\max}(\mathbf{L}\widetilde{\mathbf{\Omega}})B}{W} > 0.$$

*In case of $B = 0$, set $W = 0$ and $\beta = \tilde{B}$. If the stepsize*

$$\gamma \leq \frac{1}{A\lambda_{\max}(\mathbf{L}\widetilde{\mathbf{\Omega}}) + W\tilde{A}},$$

*then the iterates of* VR-GradSkip+ *(Algorithm 3) satisfy*

$$\mathbb{E}\left[\Psi_t\right] \leq \left(1 - \min\{\gamma\mu, \delta, \beta\}\right)^t \Psi_0 + \gamma^2\frac{\lambda_{\max}(\mathbf{L}\widetilde{\mathbf{\Omega}})C + W\tilde{C}}{\min\{\gamma\mu, \delta, \beta\}},$$

*where*

$$\delta = 1 - \frac{1}{1 + \lambda_{\min}(\mathbf{\Omega})}\left(1 - \frac{1}{(1+\omega)^2}\right), \quad \widetilde{\mathbf{\Omega}} = \mathbf{I} + \omega(\omega+2)\mathbf{\Omega}(\mathbf{I}+\mathbf{\Omega})^{-1}. \tag{12}$$

## B.3 Special Cases

- GradSkip+. Consider the case when stochastic gradients are full batch gradients, i.e., $g_t = \nabla f(x_t)$ for all $t \geq 0$. Then Algorithm 3 reduces to GradSkip+.

- ProxSkip-VR. To recover ProxSkip-VR from VR-GradSkip+, we need the same conditions we had for recovering ProxSkip from GradSkip+. That is, let $\mathcal{C}_\mathbf{\Omega}$ be the identity compressor (i.e., $\mathbf{\Omega} = \mathbf{I}$) and $\mathcal{C}_\omega$ be the Bernoulli compressor $\mathcal{C}_p$ with parameter $p \in (0, 1]$ (note that here $\omega = 1/p - 1$). In this case, $\hat{h}_{t+1} \equiv h_t$ and

$$x_{t+1} = \begin{cases} \text{prox}_{\frac{\gamma}{p}\psi}\left(\hat{x}_{t+1} - \frac{\gamma}{p}h_t\right), & \text{with probability } p, \\ \hat{x}_{t+1}, & \text{with probability } 1-p. \end{cases}$$

Thus, we recover the ProxSkip-VR algorithm.

### B.4  Proof of Theorem B.2

Here, we start proving the convergence of Algorithm 3 by first proving some auxiliary lemmas. Let

$$w_t := x_t - \gamma g_t, \quad \text{and} \quad w_\star := x_\star - \gamma \nabla f(x_\star).$$

**Lemma B.3** (Proof in Appendix B.5.1). *If $\gamma > 0$ and $\mathcal{C}_\omega \in \mathbb{B}^d(\omega)$, $\mathcal{C}_{\mathbf{\Omega}} \in \mathbb{B}^d(\mathbf{\Omega})$, then*

$$
\begin{aligned}
\mathbb{E}_t \left[ \Psi_{t+1} - \gamma^2 W \sigma_{t+1} \mid g_t \right] \quad \leq \quad & \|w_t - w_\star\|^2 \\
& + \left( 1 - \frac{1}{(1+\omega)^2} \right) \gamma^2 (1+\omega)^2 \|g_t - h_\star\|^2_{\mathbf{I} - (\mathbf{I} + \mathbf{\Omega})^{-1}} \\
& + \left( 1 - \frac{1}{(1+\omega)^2} \right) \gamma^2 (1+\omega)^2 \|h_t - h_\star\|^2_{(\mathbf{I} + \mathbf{\Omega})^{-1}},
\end{aligned}
$$

*where the expectation is with respect to the randomness from $\mathcal{C}_\omega$ and $\mathcal{C}_{\mathbf{\Omega}}$.*

Next, we upper bound the first two terms.

**Lemma B.4** (Proof in Appendix B.5.2). *Denote $\widetilde{\mathbf{\Omega}} = \mathbf{I} + \omega(\omega + 2)\mathbf{\Omega}(\mathbf{I} + \mathbf{\Omega})^{-1}$. Then*

$$
\begin{aligned}
& \mathbb{E}_t \left[ \|w_t - w_\star\|^2 \right] + \left( 1 - \frac{1}{(1+\omega)^2} \right)(1+\omega)^2 \gamma^2 \mathbb{E}_t \left[ \|g_t - h_\star\|^2_{\mathbf{I} - (\mathbf{I} + \mathbf{\Omega})^{-1}} \right] \\
& \leq (1 - \gamma\mu) \|x_t - x_\star\|^2 - 2\gamma \left( 1 - \gamma A \lambda_{\max}(\mathbf{L}\widetilde{\mathbf{\Omega}}) \right) D_f(x_t, x_\star) + \gamma^2 \lambda_{\max}(\mathbf{L}\widetilde{\mathbf{\Omega}}) B \sigma_t \\
& \quad + \gamma^2 \lambda_{\max}(\mathbf{L}\widetilde{\mathbf{\Omega}}) C.
\end{aligned}
$$

We are ready to prove the theorem.

*Proof of Theorem B.2.* The proof is a direct combination of the two lemmas.

$$
\begin{aligned}
\mathbb{E}\left[\Psi_{t+1}\right] \quad \leq \quad & (1 - \gamma\mu) \|x_t - x_\star\|^2 - 2\gamma \left( 1 - \gamma A \lambda_{\max}(\mathbf{L}\widetilde{\mathbf{\Omega}}) \right) D_f(x_t, x_\star) \\
& + \gamma^2 \lambda_{\max}(\mathbf{L}\widetilde{\mathbf{\Omega}}) B \sigma_t + \gamma^2 \lambda_{\max}(\mathbf{L}\widetilde{\mathbf{\Omega}}) C \\
& + \left( 1 - \frac{1}{(1+\omega)^2} \right) \gamma^2 (1+\omega)^2 \|h_t - h_\star\|^2_{(\mathbf{I} + \mathbf{\Omega})^{-1}} \\
& + \gamma^2 W \left( 2\tilde{A} D_f(x_t, x_\star) + \tilde{B} \sigma_t + \tilde{C} \right) \\
= \quad & (1 - \gamma\mu) \|x_t - x_\star\|^2 - 2\gamma \left( 1 - \gamma(A \lambda_{\max}(\mathbf{L}\widetilde{\mathbf{\Omega}}) + W\tilde{A}) \right) D_f(x_t, x_\star) \\
& + \frac{\omega(\omega + 2)}{(1 + \lambda_{\min}(\mathbf{\Omega}))(1+\omega)^2} \gamma^2 (1+\omega)^2 \|h_t - h_\star\|^2 \\
& + \left( \frac{\lambda_{\max}(\mathbf{L}\widetilde{\mathbf{\Omega}}) B}{W} + \tilde{B} \right) \gamma^2 W \sigma_t + \gamma^2 (\lambda_{\max}(\mathbf{L}\widetilde{\mathbf{\Omega}}) C + W\tilde{C}).
\end{aligned}
$$

Next we choose the stepsize

$$\gamma \leq \frac{1}{A \lambda_{\max}(\mathbf{L}\widetilde{\mathbf{\Omega}}) + W\tilde{A}}$$

so that the term with $D_f(x_t, x_\star)$ is non-negative and can be suppressed for further steps. Let

$$\delta = 1 - \frac{\omega(\omega + 2)}{(1 + \lambda_{\min}(\mathbf{\Omega}))(\omega + 1)^2} = 1 - \frac{1}{1 + \lambda_{\min}(\mathbf{\Omega})} \left( 1 - \frac{1}{(1+\omega)^2} \right) \in [0, 1],$$

$$\beta = 1 - \tilde{B} - \frac{\lambda_{\max}(\mathbf{L}\widetilde{\mathbf{\Omega}}) B}{W} > 0,$$

provided that $W > \frac{\lambda_{\max}(\mathbf{L}\widetilde{\mathbf{\Omega}})B}{1 - \tilde{B}}$, and continue the above derivation

$$
\begin{aligned}
\mathbb{E}\left[\Psi_{t+1}\right] &\leq \max\left\{1 - \gamma\mu, 1 - \delta, 1 - \beta\right\}\Psi_t + \gamma^2(\lambda_{\max}(\mathbf{L}\widetilde{\mathbf{\Omega}})C + W\tilde{C}) \\
&= (1 - \min\left\{\gamma\mu, \delta, \beta\right\})\Psi_t + \gamma^2(\lambda_{\max}(\mathbf{L}\widetilde{\mathbf{\Omega}})C + W\tilde{C}) \\
&\leq (1 - \min\left\{\gamma\mu, \delta, \beta\right\})^{t+1}\Psi_0 + \gamma^2\frac{\lambda_{\max}(\mathbf{L}\widetilde{\mathbf{\Omega}})C + W\tilde{C}}{\min\left\{\gamma\mu, \delta, \beta\right\}}.
\end{aligned}
$$

$\square$

### B.5 Proof of Auxiliary Lemmas

#### B.5.1 Proof of Lemma B.3

**Lemma B.3.** *If $\gamma > 0$ and $\mathcal{C}_\omega \in \mathbb{B}^d(\omega)$, $\mathcal{C}_{\mathbf{\Omega}} \in \mathbb{B}^d(\mathbf{\Omega})$, then*

$$
\begin{aligned}
\mathbb{E}_t\left[\Psi_{t+1} - \gamma^2 W\sigma_{t+1} \mid g_t\right] &\leq \|w_t - w_\star\|^2 \\
&+ \left(1 - \frac{1}{(1+\omega)^2}\right)\gamma^2(1+\omega)^2\|g_t - h_\star\|_{\mathbf{I} - (\mathbf{I}+\mathbf{\Omega})^{-1}}^2 \\
&+ \left(1 - \frac{1}{(1+\omega)^2}\right)\gamma^2(1+\omega)^2\|h_t - h_\star\|_{(\mathbf{I}+\mathbf{\Omega})^{-1}}^2,
\end{aligned}
$$

*where the expectation is with respect to the randomness from $\mathcal{C}_\omega$ and $\mathcal{C}_{\mathbf{\Omega}}$.*

*Proof.* In order to simplify notation, let $P(\cdot) := \mathrm{prox}_{\gamma(1+\omega)\psi}(\cdot)$, and

$$
x := \hat{x}_{t+1} - \gamma(1+\omega)\hat{h}_{t+1}, \qquad y := x_\star - \gamma(1+\omega)h_\star. \tag{13}
$$

**STEP 1 (Optimality conditions).** Using the first-order optimality conditions for $f + \psi$ and using $h_\star := \nabla f(x_\star)$, we obtain the following fixed-point identity for $x_\star$:

$$
x_\star = \mathrm{prox}_{\gamma(1+\omega)\psi}(x_\star - \gamma(1+\omega)h_\star) \overset{(13)}{=} P(y). \tag{14}
$$

**STEP 2 (Recalling the steps of the method).** Recall that the vectors $x_{t+1}$ and $h_{t+1}$ are in Algorithm 3 updated as follows:

$$
x_{t+1} = \hat{x}_{t+1} - \gamma\hat{g}_t = \hat{x}_{t+1} - \frac{1}{1+\omega}\mathcal{C}_\omega\left(\hat{x}_{t+1} - P(x)\right), \tag{15}
$$

and

$$
h_{t+1} = \hat{h}_{t+1} + \frac{1}{\gamma(1+\omega)}(x_{t+1} - \hat{x}_{t+1}) = \hat{h}_{t+1} - \frac{1}{\gamma(1+\omega)^2}\mathcal{C}_\omega\left(\hat{x}_{t+1} - P(x)\right). \tag{16}
$$

**STEP 3 (One-step expectation of the Lyapunov function).** The expected value of the Lyapunov function

$$
\Psi_t := \|x_t - x_\star\|^2 + \gamma^2(1+\omega)^2\|h_t - h_\star\|^2 + \gamma^2 W\sigma_t \tag{17}
$$

at time $t + 1$, with respect to the randomness of $\mathcal{C}_\omega$, is

$$\mathbb{E}_t \left[ \Psi_{t+1} - \gamma^2 W \sigma_{t+1} \mid \mathcal{C}_\Omega, g_t \right]$$

$$= \mathbb{E}_t \left[ \left\| \hat{x}_{t+1} - \frac{1}{1+\omega} \mathcal{C}_\omega \left( \hat{x}_{t+1} - P(x) \right) - x_\star \right\|^2 \mid \mathcal{C}_\Omega, g_t \right]$$

$$+ \ \mathbb{E}_t \left[ \gamma^2 (1+\omega)^2 \left\| \hat{h}_{t+1} - \frac{1}{\gamma(1+\omega)^2} \mathcal{C}_\omega \left( \hat{x}_{t+1} - P(x) \right) - h_\star \right\|^2 \mid \mathcal{C}_\Omega, g_t \right]$$

$$= \mathbb{E}_t \left[ \| \hat{x}_{t+1} - x_\star \|^2 - \frac{2}{1+\omega} \left\langle \mathcal{C}_\omega \left( \hat{x}_{t+1} - P(x) \right), \hat{x}_{t+1} - x_\star \right\rangle \right.$$

$$+ \ \frac{1}{(1+\omega)^2} \| \mathcal{C}_\omega \left( \hat{x}_{t+1} - P(x) \right) \|^2 \mid \mathcal{C}_\Omega, g_t \Big]$$

$$+ \mathbb{E}_t \left[ \gamma^2 (1+\omega)^2 \left\| \hat{h}_{t+1} - h_\star \right\|^2 - 2\gamma \left\langle \mathcal{C}_\omega \left( \hat{x}_{t+1} - P(x) \right), \hat{h}_{t+1} - h_\star \right\rangle \right.$$

$$+ \ \frac{1}{(1+\omega)^2} \| \mathcal{C}_\omega \left( \hat{x}_{t+1} - P(x) \right) \|^2 \mid \mathcal{C}_\Omega, g_t \Big]$$

$$\leq \| \hat{x}_{t+1} - x_\star \|^2 + \frac{2}{1+\omega} \left\langle P(x) - \hat{x}_{t+1}, \hat{x}_{t+1} - x_\star \right\rangle + \frac{1}{1+\omega} \| P(x) - \hat{x}_{t+1} \|^2$$

$$+ \gamma^2 (1+\omega)^2 \left\| \hat{h}_{t+1} - h_\star \right\|^2 + \frac{2}{1+\omega} \left\langle P(x) - \hat{x}_{t+1}, \gamma(1+\omega)(\hat{h}_{t+1} - h_\star) \right\rangle$$

$$+ \ \frac{1}{1+\omega} \| P(x) - \hat{x}_{t+1} \|^2$$

$$= \| \hat{x}_{t+1} - x_\star \|^2 + \frac{1}{1+\omega} \left( \| P(x) - x_\star \|^2 - \| \hat{x}_{t+1} - x_\star \|^2 \right)$$

$$+ \gamma^2 (1+\omega)^2 \left\| \hat{h}_{t+1} - h_\star \right\|^2$$

$$+ \ \frac{1}{1+\omega} \left( \left\| P(x) - \hat{x}_{t+1} + \gamma(1+\omega)(\hat{h}_{t+1} - h_\star) \right\|^2 - \gamma^2 (1+\omega)^2 \left\| \hat{h}_{t+1} - h_\star \right\|^2 \right)$$

$$= \left( 1 - \frac{1}{1+\omega} \right) \left( \| \hat{x}_{t+1} - x_\star \|^2 + \gamma^2 (1+\omega)^2 \left\| \hat{h}_{t+1} - h_\star \right\|^2 \right)$$

$$+ \ \frac{1}{1+\omega} \left( \| P(x) - x_\star \|^2 + \left\| P(x) - \hat{x}_{t+1} + \gamma(1+\omega)(\hat{h}_{t+1} - h_\star) \right\|^2 \right)$$

$$= \left( 1 - \frac{1}{1+\omega} \right) \left( \| \hat{x}_{t+1} - x_\star \|^2 + \gamma^2 (1+\omega)^2 \left\| \hat{h}_{t+1} - h_\star \right\|^2 \right)$$

$$+ \ \frac{1}{1+\omega} \left( \| P(x) - P(y) \|^2 + \| P(x) - x + y - P(y) \|^2 \right).$$

**STEP 4 (Applying firm non-expansiveness).** Applying firm non-expansiveness of prox operator $P$, this leads to the inequality

$$\mathbb{E}_t \left[ \Psi_{t+1} - \gamma^2 W \sigma_{t+1} \mid \mathcal{C}_\Omega, g_t \right]$$

$$\leq \left( 1 - \frac{1}{1+\omega} \right) \left( \| \hat{x}_{t+1} - x_\star \|^2 + \gamma^2 (1+\omega)^2 \left\| \hat{h}_{t+1} - h_\star \right\|^2 \right)$$

$$+ \ \frac{1}{1+\omega} \| x - y \|^2$$

$$= \left( 1 - \frac{1}{1+\omega} \right) \left( \| \hat{x}_{t+1} - x_\star \|^2 + \gamma^2 (1+\omega)^2 \left\| \hat{h}_{t+1} - h_\star \right\|^2 \right)$$

$$+ \ \frac{1}{1+\omega} \left\| \hat{x}_{t+1} - \gamma(1+\omega)\hat{h}_{t+1} - (x_\star - \gamma(1+\omega)h_\star) \right\|^2$$

$$= \left( 1 - \frac{1}{1+\omega} \right) \left( \| \hat{x}_{t+1} - x_\star \|^2 + \gamma^2 (1+\omega)^2 \left\| \hat{h}_{t+1} - h_\star \right\|^2 \right)$$

$$+ \ \frac{1}{1+\omega} \left\| \hat{x}_{t+1} - x_\star - \gamma(1+\omega) \left( \hat{h}_{t+1} - h_\star \right) \right\|^2.$$

**STEP 5 (Simple algebra).** Next, we expand the squared norm and collect the terms, obtaining

$$\mathbb{E}_t\left[\Psi_{t+1} - \gamma^2 W\sigma_{t+1} \mid \mathcal{C}_{\boldsymbol{\Omega}}, g_t\right]$$

$$\leq \left(1 - \frac{1}{1+\omega}\right)\left(\|\hat{x}_{t+1} - x_\star\|^2 + \gamma^2(1+\omega)^2\left\|\hat{h}_{t+1} - h_\star\right\|^2\right)$$

$$+ \frac{1}{1+\omega}\|\hat{x}_{t+1} - x_\star\|^2 - 2\gamma\langle\hat{x}_{t+1} - x_\star, \hat{h}_{t+1} - h_\star\rangle + \gamma^2(1+\omega)\|\hat{h}_{t+1} - h_\star\|^2$$

$$= \|\hat{x}_{t+1} - x_\star\|^2 - 2\gamma\langle\hat{x}_{t+1} - x_\star, \hat{h}_{t+1} - h_\star\rangle + \gamma^2(1+\omega)^2\|\hat{h}_{t+1} - h_\star\|^2$$

$$= \|\hat{x}_{t+1} - x_\star - \gamma(\hat{h}_{t+1} - h_\star)\|^2 - \gamma^2\|\hat{h}_{t+1} - h_\star\|^2 + \gamma^2(1+\omega)^2\|\hat{h}_{t+1} - h_\star\|^2$$

$$= \|\hat{x}_{t+1} - x_\star - \gamma(\hat{h}_{t+1} - h_\star)\|^2 + \left(1 - \frac{1}{(1+\omega)^2}\right)\gamma^2(1+\omega)^2\|\hat{h}_{t+1} - h_\star\|^2.$$

**STEP 6 (Tower property).** Applying the expectation with respect to the randomness of $\mathcal{C}_{\boldsymbol{\Omega}}$ and using the tower property, we get

$$\mathbb{E}_t\left[\Psi_{t+1} - \gamma^2 W\sigma_{t+1} \mid g_t\right]$$

$$= \mathbb{E}_t\left[\left\|x_t - \gamma(g_t - \hat{h}_{t+1}) - x_\star - \gamma(\hat{h}_{t+1} - h_\star)\right\|^2 \mid g_t\right]$$

$$+ \left(1 - \frac{1}{(1+\omega)^2}\right)\gamma^2(1+\omega)^2\mathbb{E}_t\left[\left\|g_t - (\mathbf{I}+\boldsymbol{\Omega})^{-1}\mathcal{C}_{\boldsymbol{\Omega}}(g_t - h_t) - h_\star\right\|^2 \mid g_t\right]$$

$$= \|x_t - \gamma g_t - (x_\star - \gamma h_\star)\|^2$$

$$+ \left(1 - \frac{1}{(1+\omega)^2}\right)\gamma^2(1+\omega)^2\mathbb{E}_t\left[\left\|g_t - h_\star - (\mathbf{I}+\boldsymbol{\Omega})^{-1}\mathcal{C}_{\boldsymbol{\Omega}}(g_t - h_t)\right\|^2 \mid g_t\right]$$

$$\leq \|w_t - w_\star\|^2 + \left(1 - \frac{1}{(1+\omega)^2}\right)\gamma^2(1+\omega)^2\|g_t - h_\star\|^2$$

$$+ \left(1 - \frac{1}{(1+\omega)^2}\right)\gamma^2(1+\omega)^2\left(2\langle g_t - h_\star, h_t - g_t\rangle_{(\mathbf{I}+\boldsymbol{\Omega})^{-1}} + \|g_t - h_t\|^2_{(\mathbf{I}+\boldsymbol{\Omega})^{-1}}\right)$$

$$= \|w_t - w_\star\|^2 + \left(1 - \frac{1}{(1+\omega)^2}\right)\gamma^2(1+\omega_2)^2\|g_t - h_\star\|^2$$

$$+ \left(1 - \frac{1}{(1+\omega)^2}\right)\gamma^2(1+\omega_2)^2\left(\|h_t - h_\star\|^2_{(\mathbf{I}+\boldsymbol{\Omega})^{-1}} - \|g_t - h_\star\|^2_{(\mathbf{I}+\boldsymbol{\Omega})^{-1}}\right)$$

$$= \|w_t - w_\star\|^2 + \left(1 - \frac{1}{(1+\omega)^2}\right)\gamma^2(1+\omega)^2\|g_t - h_\star\|^2_{\mathbf{I}-(\mathbf{I}+\boldsymbol{\Omega})^{-1}}$$

$$+ \left(1 - \frac{1}{(1+\omega)^2}\right)\gamma^2(1+\omega)^2\|h_t - h_\star\|^2_{(\mathbf{I}+\boldsymbol{\Omega})^{-1}}.$$

$\square$

### B.5.2 Proof of Lemma B.4

**Lemma B.4.** *Denote* $\widetilde{\boldsymbol{\Omega}} = \mathbf{I} + \omega(\omega+2)\boldsymbol{\Omega}(\mathbf{I}+\boldsymbol{\Omega})^{-1}$. *Then*

$$\mathbb{E}_t\left[\|w_t - w_\star\|^2\right] + \left(1 - \frac{1}{(1+\omega)^2}\right)(1+\omega)^2\gamma^2\mathbb{E}_t\left[\|g_t - h_\star\|^2_{\mathbf{I}-(\mathbf{I}+\boldsymbol{\Omega})^{-1}}\right]$$

$$\leq (1 - \gamma\mu)\|x_t - x_\star\|^2 - 2\gamma\left(1 - \gamma A\lambda_{\max}(\mathbf{L}\widetilde{\boldsymbol{\Omega}})\right)D_f(x_t, x_\star) + \gamma^2\lambda_{\max}(\mathbf{L}\widetilde{\boldsymbol{\Omega}})B\sigma_t$$

$$+ \gamma^2\lambda_{\max}(\mathbf{L}\widetilde{\boldsymbol{\Omega}})C.$$

*Proof.* Expanding the first term and rearranging terms, we get

$$
\begin{aligned}
\mathbb{E}_t &\left[\|w_t - w_\star\|^2\right] + \left(1 - \frac{1}{(1+\omega)^2}\right)(1+\omega)^2\gamma^2\mathbb{E}_t\left[\|g_t - h_\star\|^2_{\mathbf{I}-(\mathbf{I}+\mathbf{\Omega})^{-1}}\right] \\
&= \mathbb{E}_t\left[\|x_t - x_\star - \gamma(g_t - \nabla f(x_\star))\|^2\right] + \omega(\omega+2)\gamma^2\mathbb{E}_t\left[\|g_t - \nabla f(x_\star)\|^2_{\mathbf{\Omega}(\mathbf{I}+\mathbf{\Omega})^{-1}}\right] \\
&= \|x_t - x_\star\|^2 - 2\gamma\langle x_t - x_\star, \nabla f(x_t) - \nabla f(x_\star)\rangle \\
&\quad + \gamma^2\mathbb{E}_t\left[\|g_t - \nabla f(x_\star)\|^2\right] + \omega(\omega+2)\gamma^2\mathbb{E}_t\left[\|g_t - \nabla f(x_\star)\|^2_{\mathbf{\Omega}(\mathbf{I}+\mathbf{\Omega})^{-1}}\right] \\
&\leq (1-\gamma\mu)\|x_t - x_\star\|^2 - 2\gamma D_f(x_t, x_\star) + \gamma^2\mathbb{E}_t\left[\|g_t - \nabla f(x_\star)\|^2_{\widetilde{\mathbf{\Omega}}}\right] \\
&\leq (1-\gamma\mu)\|x_t - x_\star\|^2 - 2\gamma D_f(x_t, x_\star) + \gamma^2\lambda_{\max}(\mathbf{L}\widetilde{\mathbf{\Omega}})\mathbb{E}_t\left[\|g_t - \nabla f(x_\star)\|^2_{\mathbf{L}^{-1}}\right] \\
&\leq (1-\gamma\mu)\|x_t - x_\star\|^2 - 2\gamma D_f(x_t, x_\star) + \gamma^2\lambda_{\max}(\mathbf{L}\widetilde{\mathbf{\Omega}})(2AD_f(x_t, x_\star) + B\sigma_t + C) \\
&= (1-\gamma\mu)\|x_t - x_\star\|^2 - 2\gamma\left(1 - \gamma A\lambda_{\max}(\mathbf{L}\widetilde{\mathbf{\Omega}})\right)D_f(x_t, x_\star) + \gamma^2\lambda_{\max}(\mathbf{L}\widetilde{\mathbf{\Omega}})B\sigma_t \\
&\quad + \gamma^2\lambda_{\max}(\mathbf{L}\widetilde{\mathbf{\Omega}})C.
\end{aligned}
$$

$\square$

# C    Proofs for Section 3 (GradSkip)

## C.1    Proof of Lemma 3.1

**Lemma 3.1** (Fake local steps). *Suppose that Algorithm 1 does not communicate for $\tau \geq 1$ consecutive iterates, i.e., $\theta_t = \theta_{t+1} = \cdots = \theta_{t+\tau-1} = 0$ for some fixed $t \geq 0$. Besides, let for some client $i \in [n]$ we have $\eta_{i,t} = 0$. Then, regardless of the coin tosses $\{\eta_{i,t+j}\}_{j=1}^{\tau}$, client $i$ does fake local steps without any gradient computation in $\tau$ iterates. Formally, for all $j = 1, 2, \ldots, \tau + 1$, we have*

$$
\begin{aligned}
\hat{x}_{i,t+j} &= x_{i,t+j} = x_{i,t}, \\
\hat{h}_{i,t+j} &= h_{i,t+j} = h_{i,t} = \nabla f_i(x_{i,t}).
\end{aligned}
\tag{18}
$$

*Proof.* The proof is rather straightforward and follows by following the corresponding lines of the algorithm. Note that $\eta_{i,t} = \theta_t = 0$ implies (see lines 6 and 7 in Algorithm 1) that

$$
\hat{x}_{i,t+1} = x_{i,t+1} = x_{i,t}, \tag{19}
$$

$$
\hat{h}_{i,t+1} = h_{i,t+1} = h_{i,t} = \nabla f_i(x_{i,t}), \tag{20}
$$

which proves (18) when $j = 1$. Consider the two possible cases for $\eta_{i,t+1}$ coupled with $\theta_{t+1} = 0$. If $\eta_{i,t+1} = 1$, then

$$
\begin{aligned}
\hat{x}_{i,t+2} &= x_{i,t+1} - \gamma(\nabla f_i(x_{i,t+1}) - h_{i,t+1}) \\
&\stackrel{(19)}{=} x_{i,t+1} - \gamma(\nabla f_i(x_{i,t}) - h_{i,t+1}) \\
&\stackrel{(20)}{=} x_{i,t+1} \\
&\stackrel{(19)}{=} x_{i,t},
\end{aligned}
$$

and

$$
\hat{h}_{i,t+2} = h_{i,t+1} \stackrel{(20)}{=} h_{i,t} = \nabla f_i(x_{i,t}).
$$

In case of $\eta_{i,t+1} = 0$, we have

$$
\hat{x}_{i,t+2} = x_{i,t+1} \stackrel{(19)}{=} x_{i,t}
$$

and

$$\hat{h}_{i,t+2} = \nabla f_i(x_{i,t+1}) \overset{(19)}{=} \nabla f_i(x_{i,t}) \overset{(19)}{=} h_{i,t}.$$

Hence, in both cases, we get

$$\hat{x}_{i,t+2} = x_{i,t+1} = x_{i,t}, \tag{21}$$

$$\hat{h}_{i,t+2} = h_{i,t} = \nabla f_i(x_{i,t}). \tag{22}$$

It remains to combine (21)–(22) with the condition that $\theta_{t+1} = 0$, which implies

$$x_{i,t+2} = \hat{x}_{i,t+2}, \quad h_{i,t+2} = \hat{h}_{i,t+2}.$$

Thus, we proved (18) when $j = 2$. The proof can be completed by applying induction on $j$. $\qquad\square$

## C.2   Proof of Lemma 3.2

**Lemma 3.2** (Expected number of local steps). *The expected number of local gradient computations in each communication round of* GradSkip *is* $1/(1-q_i(1-p))$ *for all clients* $i \in [n]$.

*Proof.* As mentioned in the text preceding the lemma, the proof follows from the fact that for two geometric random variables $\Theta \sim \mathrm{Geo}(p)$ and $H \sim \mathrm{Geo}(q)$, their minimum $\min\{\Theta, H\}$ is also a geometric random variable with parameter $1 - (1-p)(1-q)$. To see this, consider the corresponding Bernoulli trials with success probability $p$ and $q$ for each geometric random variable. Notice that the probability that both trials fail is $(1-p)(1-q)$. Hence, $\min\{\Theta, H\}$ is the number of joint trials of the two Bernoulli variables until one of them succeeds with probability $1 - (1-p)(1-q)$. Therefore, $\min\{\Theta, H\}$ is also a geometric random variable with success probability $1 - (1-p)(1-q)$. $\qquad\square$

## C.3   Proof of Theorem 3.5

**Theorem 3.5.** *Let Assumption 3.4 hold. If the stepsize satisfies*

$$\gamma \leq \min_i \left\{ \frac{1}{L_i} \frac{p^2}{1 - q_i\left(1 - p^2\right)} \right\}$$

*and probabilities are chosen so that* $0 < p, q_i \leq 1$, *then the iterates of* GradSkip *(Algorithm 1) satisfy*

$$\mathbb{E}\left[\Psi_t\right] \leq (1 - \rho)^t \Psi_0,$$

*for all* $t \geq 1$ *with* $\rho := \min\left\{\gamma\mu, 1 - q_{max}(1 - p^2)\right\} > 0$.

We use the following two auxiliary lemmas to prove the theorem.

Denote $\mathbb{E}_t\left[\,\cdot\,\right] := \mathbb{E}\left[\,\cdot \mid x_{1,t}, \cdots, x_{n,t}\right]$ the conditional expectation with respect to the randomness of all local models $x_{1,t}, \cdots, x_{n,t}$ at $t^{th}$ iterate.

**Lemma C.1** (Proof in Appendix C.4.1). *If* $\gamma > 0$ *and* $0 \leq p, q_i \leq 1$, *then*

$$\mathbb{E}_t\left[\Psi_{t+1}\right] = \sum_{i=1}^{n} \Bigg[ \|w_{i,t} - w_{i,\star}\|^2 + (1 - q_i)\left(1 - p^2\right)\frac{\gamma^2}{p^2}\|\nabla f(x_{i,t}) - h_{i,\star}\|^2$$

$$+ q_i\left(1 - p^2\right)\frac{\gamma^2}{p^2}\|h_{i,t} - h_{i,\star}\|^2 \Bigg],$$

*where the expectation is taken over* $\theta_t$ *and* $\eta_{i,t}$ *in Algorithm 1.*

Next, we upper bound the first two terms of the above equality by adjusting the stepsize.

**Lemma C.2** (Proof in Appendix C.4.2). *If*

$$0 < \gamma \leq \min_i \left\{ \frac{1}{L_i} \frac{p^2}{1 - q_i (1 - p^2)} \right\},$$

*then*

$$\|w_{i,t} - w_{i,\star}\|^2 + (1 - q_i)(1 - p^2) \frac{\gamma^2}{p^2} \|\nabla f(x_{i,t}) - h_{i,\star}\|^2 \leq (1 - \gamma\mu)\|x_{i,t} - x_\star\|^2.$$

*Proof of Theorem 3.5.* The proof of the theorem is direct combination of the above lemmas.

$$
\begin{aligned}
\mathbb{E}_t [\Psi_{t+1}] &= \sum_{i=1}^n \Bigg[ \|x_{i,t} - x_\star - \gamma (\nabla f_i(x_{i,t}) - h_{i,\star})\|^2 \\
&\qquad + (1 - q_i)(1 - p^2) \frac{\gamma^2}{p^2} \|\nabla f(x_{i,t}) - h_{i,\star}\|^2 \\
&\qquad + q_i (1 - p^2) \frac{\gamma^2}{p^2} \|h_{i,t} - h_{i,\star}\|^2 \Bigg] \\
&\leq \sum_{i=1}^n \left[ (1 - \gamma\mu) \|x_{i,t} - x_\star\|^2 + q_i (1 - p^2) \frac{\gamma^2}{p^2} \|h_{i,t} - h_{i,\star}\|^2 \right] \\
&\leq (1 - \gamma\mu) \sum_{i=1}^n \|x_{i,t} - x_\star\|^2 + q_{max} (1 - p^2) \frac{\gamma^2}{p^2} \sum_{i=1}^n \|h_{i,t} - h_{i,\star}\|^2 \\
&\leq \max \left\{ 1 - \gamma\mu, q_{max} (1 - p^2) \right\} \Psi_t \\
&= \left( 1 - \min \left\{ \gamma\mu, 1 - q_{max} (1 - p^2) \right\} \right) \Psi_t.
\end{aligned}
$$

$\square$

## C.4  Proof of Auxiliary Lemmas

### C.4.1  Proof of Lemma C.1

**Lemma C.1.** *If $\gamma > 0$ and $0 \leq p, q_i \leq 1$, then*

$$
\begin{aligned}
\mathbb{E}_t [\Psi_{t+1}] &= \sum_{i=1}^n \Bigg[ \|w_{i,t} - w_{i,\star}\|^2 + (1 - q_i)(1 - p^2) \frac{\gamma^2}{p^2} \|\nabla f(x_{i,t}) - h_{i,\star}\|^2 \\
&\qquad + q_i (1 - p^2) \frac{\gamma^2}{p^2} \|h_{i,t} - h_{i,\star}\|^2 \Bigg],
\end{aligned}
$$

*where the expectation is taken over $\theta_t$ and $\eta_{i,t}$ in Algorithm 1.*

*Proof.* In order to simplify notation, denote

$$x_i := \hat{x}_{i,t+1} - \frac{\gamma}{p}\hat{h}_{i,t+1}, \qquad y_i := x_\star - \frac{\gamma}{p}h_{i,\star}. \tag{23}$$

$$\bar{x} := \frac{1}{n} \sum_{i=1}^n x_i, \qquad \bar{y} := \frac{1}{n} \sum_{i=1}^n y_i = x_*. \tag{24}$$

**STEP 1 (Recalling the steps of the method).** Recall that

$$x_{i,t+1} = \begin{cases} \bar{x}, & \text{with probability} \quad p, \\ \hat{x}_{i,t+1}, & \text{with probability} \quad 1 - p, \end{cases} \tag{25}$$

and

$$h_{i,t+1} = \begin{cases} \hat{h}_{i,t+1} + \frac{p}{\gamma}(\bar{x} - \hat{x}_{i,t+1}), & \text{with probability} \quad p, \\ \hat{h}_{i,t+1}, & \text{with probability} \quad 1-p. \end{cases} \tag{26}$$

**STEP 2 (One-step expectation w.r.t. the global coin toss $\theta_t$).** The expected value of the Lyapunov function

$$\Psi_t := \sum_{i=1}^{n} \|x_{i,t} - x_\star\|^2 + \frac{\gamma^2}{p^2} \sum_{i=1}^{n} \|h_{i,t} - h_{i,\star}\|^2 \tag{27}$$

at $(t+1)^{th}$ iterate with respect to the coin toss $\theta_t$ is

$$
\begin{aligned}
\mathbb{E}_t & \left[\Psi_{t+1} \mid \eta_{1,t}, \ldots, \eta_{n,t}\right] \\
& \overset{(25)-(27)}{=} p \sum_{i=1}^{n} \left( \|\bar{x} - x_\star\|^2 + \frac{\gamma^2}{p^2} \left\| \hat{h}_{i,t+1} + \frac{p}{\gamma}(\bar{x} - \hat{x}_{i,t+1}) - h_{i,\star} \right\|^2 \right) \\
& \quad + (1-p) \sum_{i=1}^{n} \left( \|\hat{x}_{i,t+1} - x_\star\|^2 + \frac{\gamma^2}{p^2}\|\hat{h}_{i,t+1} - h_{i,\star}\|^2 \right) \\
& \overset{(24)}{=} p \sum_{i=1}^{n} \left( \|\bar{x} - \bar{y}\|^2 + \|\bar{x} - x_i + y_i - \bar{y}\|^2 \right) \\
& \quad + (1-p) \sum_{i=1}^{n} \left( \|\hat{x}_{i,t+1} - x_\star\|^2 + \frac{\gamma^2}{p^2}\|\hat{h}_{i,t+1} - h_{i,\star}\|^2 \right) \\
& = p \sum_{i=1}^{n} \|x_i - y_i\|^2 + (1-p) \sum_{i=1}^{n} \left( \|\hat{x}_{i,t+1} - x_\star\|^2 + \frac{\gamma^2}{p^2}\|\hat{h}_{i,t+1} - h_{i,\star}\|^2 \right) \\
& = \sum_{i=1}^{n} \left[ p \left\| \hat{x}_{i,t+1} - \frac{\gamma}{p}\hat{h}_{i,t+1} - \left(x_\star - \frac{\gamma}{p}h_{i,\star}\right) \right\|^2 \right. \\
& \quad \left. + (1-p) \left( \|\hat{x}_{i,t+1} - x_\star\|^2 + \frac{\gamma^2}{p^2}\|\hat{h}_{i,t+1} - h_{i,\star}\|^2 \right) \right].
\end{aligned}
$$

**STEP 3 (Simple algebra).** Next, we expand the squared norm and collect the terms, obtaining

$$
\begin{aligned}
\mathbb{E}_t & \left[\Psi_{t+1} \mid \eta_{1,t}, \ldots, \eta_{n,t}\right] \\
& = \sum_{i=1}^{n} \left[ p\|\hat{x}_{i,t+1} - x_\star\|^2 + p\frac{\gamma^2}{p^2}\|\hat{h}_{i,t+1} - h_{i,\star}\|^2 - 2\gamma\langle \hat{x}_{i,t+1} - x_\star, \hat{h}_{i,t+1} - h_{i,\star}\rangle \right. \\
& \quad \left. + (1-p) \left( \|\hat{x}_{i,t+1} - x_\star\|^2 + \frac{\gamma^2}{p^2}\|\hat{h}_{i,t+1} - h_{i,\star}\|^2 \right) \right] \\
& = \sum_{i=1}^{n} \left[ \|\hat{x}_{i,t+1} - x_\star\|^2 - 2\gamma\langle \hat{x}_{i,t+1} - x_\star, \hat{h}_{i,t+1} - h_{i,\star}\rangle + \frac{\gamma^2}{p^2}\|\hat{h}_{i,t+1} - h_{i,\star}\|^2 \right] \\
& = \sum_{i=1}^{n} \left[ \left\| \hat{x}_{i,t+1} - x_\star - \gamma\left(\hat{h}_{i,t+1} - h_{i,\star}\right) \right\|^2 - \gamma^2 \left\|\hat{h}_{i,t+1} - h_{i,\star}\right\|^2 + \frac{\gamma^2}{p^2}\|\hat{h}_{i,t+1} - h_{i,\star}\|^2 \right] \\
& = \sum_{i=1}^{n} \left[ \left\| \hat{x}_{i,t+1} - x_\star - \gamma\left(\hat{h}_{i,t+1} - h_{i,\star}\right) \right\|^2 + (1-p^2)\frac{\gamma^2}{p^2}\|\hat{h}_{i,t+1} - h_{i,\star}\|^2 \right].
\end{aligned}
$$

**STEP 4 (One-step expectation w.r.t. local coin tosses $\eta_{i,t}$).** Applying the expectation with respect to (independent) coin tosses $\eta_{i,t}$ and using the tower property we get

$$\mathbb{E}_t\left[\Psi_{t+1}\right]$$

$$= \sum_{i=1}^{n}\left[q_i\left(\|x_{i,t}-\gamma(\nabla f_i(x_{i,t})-h_{i,t})-x_\star-\gamma\left(h_{i,t}-h_{i,\star}\right)\|^2\right.\right.$$

$$\left.+\ \left(1-p^2\right)\frac{\gamma^2}{p^2}\|h_{i,t}-h_{i,\star}\|^2\right)$$

$$\left.+\ (1-q_i)\left(\|x_{i,t}-x_\star-\gamma\left(\nabla f(x_{i,t})-h_{i,\star}\right)\|^2+\left(1-p^2\right)\frac{\gamma^2}{p^2}\|\nabla f(x_{i,t})-h_{i,\star}\|^2\right)\right]$$

$$= \sum_{i=1}^{n}\left[q_i\left(\|x_{i,t}-x_\star-\gamma\left(\nabla f_i(x_{i,t})-h_{i,\star}\right)\|^2+\left(1-p^2\right)\frac{\gamma^2}{p^2}\|h_{i,t}-h_{i,\star}\|^2\right)\right.$$

$$\left.+\ (1-q_i)\left(\|x_{i,t}-x_\star-\gamma\left(\nabla f(x_{i,t})-h_{i,\star}\right)\|^2+\left(1-p^2\right)\frac{\gamma^2}{p^2}\|\nabla f(x_{i,t})-h_{i,\star}\|^2\right)\right]$$

$$= \sum_{i=1}^{n}\left[\|x_{i,t}-x_\star-\gamma\left(\nabla f_i(x_{i,t})-h_{i,\star}\right)\|^2+(1-q_i)\left(1-p^2\right)\frac{\gamma^2}{p^2}\|\nabla f(x_{i,t})-h_{i,\star}\|^2\right.$$

$$\left.+\ q_i\left(1-p^2\right)\frac{\gamma^2}{p^2}\|h_{i,t}-h_{i,\star}\|^2\right]$$

$$= \sum_{i=1}^{n}\left[\|w_{i,t}-w_{i,\star}\|^2+(1-q_i)\left(1-p^2\right)\frac{\gamma^2}{p^2}\|\nabla f(x_{i,t})-h_{i,\star}\|^2\right.$$

$$\left.+\ q_i\left(1-p^2\right)\frac{\gamma^2}{p^2}\|h_{i,t}-h_{i,\star}\|^2\right].$$

$\square$

### C.4.2  Proof of Lemma C.2

**Lemma C.2.** *If*

$$0<\gamma\leq\min_i\left\{\frac{1}{L_i}\frac{p^2}{1-q_i\left(1-p^2\right)}\right\},$$

*then*

$$\|w_{i,t}-w_{i,\star}\|^2+(1-q_i)\left(1-p^2\right)\frac{\gamma^2}{p^2}\|\nabla f(x_{i,t})-h_{i,\star}\|^2\leq(1-\gamma\mu)\|x_{i,t}-x_\star\|^2.$$

*Proof.* After some algebraic transformations we get

$$\|w_{i,t}-w_{i,\star}\|^2+(1-q_i)\left(1-p^2\right)\frac{\gamma^2}{p^2}\|\nabla f(x_{i,t})-h_{i,\star}\|^2$$

$$= \|x_{i,t}-x_\star-\gamma\left(\nabla f_i(x_{i,t})-h_{i,\star}\right)\|^2+(1-q_i)\left(1-p^2\right)\frac{\gamma^2}{p^2}\|\nabla f(x_{i,t})-h_{i,\star}\|^2$$

$$= \|x_{i,t}-x_\star\|^2-2\gamma\left\langle x_{i,t}-x_\star,\nabla f_i(x_{i,t})-h_{i,\star}\right\rangle$$

$$+\gamma^2\|\nabla f_i(x_{i,t})-h_{i,\star}\|^2+(1-q_i)\left(1-p^2\right)\frac{\gamma^2}{p^2}\|\nabla f(x_{i,t})-h_{i,\star}\|^2$$

$$\leq (1-\gamma\mu)\|x_{i,t}-x_\star\|^2-2\gamma D_{f_i}(x_{i,t},x_\star)$$

$$+\gamma^2\left(1+\frac{(1-q_i)\left(1-p^2\right)}{p^2}\right)\|\nabla f_i(x_{i,t})-h_{i,\star}\|^2$$

$$\leq (1-\gamma\mu)\|x_{i,t}-x_\star\|^2-2\gamma D_{f_i}(x_{i,t},x_\star)\left(1-\gamma L_i\left(\frac{p^2+(1-q_i)\left(1-p^2\right)}{p^2}\right)\right)$$

$$\leq (1-\gamma\mu)\|x_{i,t}-x_\star\|^2,$$

where we used the bound

$$\|\nabla f_i(x_{i,t}) - h_{i,\star}\|^2 \leq 2L_i D_{f_i}(x_{i,t}, x_\star)$$

and the last inequality holds since

$$\gamma \leq \frac{1}{L_i} \frac{p^2}{1 - q_i(1 - p^2)}.$$

$\square$

### C.5  Proof of Theorem 3.6

**Theorem 3.6** (Optimal parameter choices)**.** *Let Assumption 3.4 hold and choose probabilities*

$$q_i = \frac{1 - \frac{1}{\kappa_i}}{1 - \frac{1}{\kappa_{\max}}} \leq 1 \quad and \quad p = \frac{1}{\sqrt{\kappa_{\max}}}.$$

*Then, with the largest admissible stepsize $\gamma = 1/L_{\max}$, GradSkip enjoys the following properties:*

*(i) $\mathcal{O}\left(\kappa_{\max} \log 1/\varepsilon\right)$ iteration complexity,*

*(ii) $\mathcal{O}\left(\sqrt{\kappa_{\max}} \log 1/\varepsilon\right)$ communication complexity,*

*(iii) for each client $i \in [n]$, the expected number of local gradient computations per communication round is*

$$\frac{1}{1 - q_i(1 - p)} = \frac{\kappa_i(1 + \sqrt{\kappa_{\max}})}{\kappa_i + \sqrt{\kappa_{\max}}} \leq \min\left\{\kappa_i, \sqrt{\kappa_{\max}}\right\}.$$

*Proof.* From the choice of $q_i = \frac{1 - 1/\kappa_i}{1 - 1/\kappa_{\max}}$, we immediately imply $q_{\max} = 1$. Furthermore, choosing the optimal $p = \frac{1}{\sqrt{\kappa_{\max}}}$, we get

$$\gamma = \min_i\left\{\frac{1}{L_i} \frac{p^2}{1 - q_i(1 - p^2)}\right\} = \min_i\left\{\frac{L_i p^2}{L_i \mu}\right\} = \frac{1}{L_{max}}.$$

Now, if we plug these values back to the rate (6), we get the best rate of ProxSkip as

$$1 - \min\left\{\gamma\mu, 1 - q_{max}(1 - p^2)\right\}\} = 1 - \min\left\{\frac{\mu}{L_{\max}}, p^2\right\} = 1 - \frac{\mu}{L_{\max}} = 1 - \frac{1}{\kappa_{\max}}.$$

This implies $\mathcal{O}\left(\kappa_{\max} \log \frac{1}{\varepsilon}\right)$ total iteration complexity of the method. Due to the choice $p = \frac{1}{\sqrt{\kappa_{\max}}}$, the method enjoys $\mathcal{O}\left(\sqrt{\kappa_{\max}} \log \frac{1}{\varepsilon}\right)$ accelerated communication complexity.

We have two geometric random variables, $\Theta \sim \text{Geom}(p)$ and $H_i \sim \text{Geom}(1 - q_i)$, for each client describing local training. From the algorithm description, we see that the number of local steps for client $i$ is $\min\{\Theta, H_i\}$, which is still a Geometric random variable with parameter $1 - q_i(1 - p)$. Therefore, the expected number of local steps for client $i$ is the inverse of that parameter, i.e., $\frac{1}{1 - q_i(1-p)}$. If we plug in the values for $p$ and $q_i$, we have

$$
\begin{aligned}
\mathbb{E}\left[\min\{\Theta, H_i\}\right] &= \frac{1}{1 - q_i(1 - p)} = \frac{1}{1 - \left(1 - \frac{1}{\sqrt{\kappa_{\max}}}\right)\frac{1 - 1/\kappa_i}{1 - 1/\kappa_{\max}}} \\
&= \frac{1}{1 - \frac{1 - 1/\kappa_i}{1 + 1/\sqrt{\kappa_{\max}}}} = \frac{1 + 1/\sqrt{\kappa_{\max}}}{1/\kappa_i + 1/\sqrt{\kappa_{\max}}} \\
&= \frac{\kappa_i(1 + \sqrt{\kappa_{\max}})}{\kappa_i + \sqrt{\kappa_{\max}}} \leq \min\left\{\kappa_i, \sqrt{\kappa_{\max}}\right\},
\end{aligned}
$$

where the last inequality can be verified with simple algebraic steps.

$\square$

# D Proofs for Section 4 (GradSkip+)

## D.1 Proof of Lemma 4.2

**Lemma 4.2.**

$$\mathbb{B}^d(\mathbf{\Omega}) \subseteq \mathbb{B}^d \left( \frac{(1 + \lambda_{\max}(\mathbf{\Omega}))^2}{(1 + \lambda_{\min}(\mathbf{\Omega}))} - 1 \right).$$

*Proof.* The proof follows from the following simple inequalities:

$$\|x\|^2_{(\mathbf{I}+\mathbf{\Omega})^{-1}} \leq \lambda_{\max}\left((\mathbf{I}+\mathbf{\Omega})^{-1}\right) \|x\|^2 = \frac{1}{1 + \lambda_{\min}(\mathbf{\Omega})} \|x\|^2,$$

$$\|(\mathbf{I}+\mathbf{\Omega})^{-1}\mathcal{C}(x)\|^2 \geq \lambda_{\min}\left((\mathbf{I}+\mathbf{\Omega})^{-1}\right)^2 \|\mathcal{C}(x)\|^2 = \frac{1}{(1 + \lambda_{\max}(\mathbf{\Omega}))^2} \|\mathcal{C}(x)\|^2.$$

$\square$

## D.2 Proof of Theorem 4.5

**Theorem 4.5.** *Let Assumption 4.4 hold, $\mathcal{C}_\omega \in \mathbb{B}^d(\omega)$ and $\mathcal{C}_{\mathbf{\Omega}} \in \mathbb{B}^d(\mathbf{\Omega})$ be the compression operators, and*

$$\widetilde{\mathbf{\Omega}} := \mathbf{I} + \omega(\omega + 2)\mathbf{\Omega}(\mathbf{I}+\mathbf{\Omega})^{-1}.$$

*Then, if the stepsize $\gamma \leq \lambda_{\max}^{-1}(\mathbf{L}\widetilde{\mathbf{\Omega}})$, the iterates of* GradSkip+ *(Algorithm 2) satisfy*

$$\mathbb{E}\left[\Psi_t\right] \leq (1 - \min\{\gamma\mu, \delta\})^t \Psi_0,$$

*where*

$$\delta = 1 - \frac{1}{1 + \lambda_{\min}(\mathbf{\Omega})} \left(1 - \frac{1}{(1+\omega)^2}\right) \in [0, 1].$$

*Proof.* Since GradSkip+ is a special case of VR-GradSkip+, Theorem 4.5 follows directly as a corollary of Theorem B.2. What remains is to verify that the gradient estimator satisfies the condition in Assumption B.1 and to identify the associated constants. The lemma below establishes this; the final step is to substitute these values into Theorem B.2. $\square$

**Lemma D.1.** *Let Assumption 4.4 hold. Then for the gradient estimator $g_t = \nabla f(x_t)$, Assumption B.1 holds with the following parameters:*

$$A = 1, \quad B = 0, \quad C = 0, \quad \tilde{A} = 0, \quad \tilde{B} = 0, \quad \tilde{C} = 0, \quad \sigma_t \equiv 0.$$

*Proof.* The proof is rather trivial and follows from the $\mathbf{L}$-smoothness of $f$,

$$\mathbb{E}\left[\|g_t - \nabla f(x_\star)\|^2_{\mathbf{L}^{-1}}\right] = \|\nabla f(x_t) - \nabla f(x_\star)\|^2_{\mathbf{L}^{-1}} \leq 2D_f(x_t, x_\star).$$

$\square$

