# OpenReview forum: "GradSkip: Communication-Accelerated Local Gradient Methods with Better Computational Complexity"
_TMLR — Accepted by TMLR_

### Review · Reviewer_wr9e · 2024-12-28

**Summary Of Contributions:**

This paper studies a class of federated learning methods. It first proposes a new method called GradSkip -- an extension the existing ProxSkip method, which considers a regularized optimization problem and uses randomization. Then, the paper proposes a generalization of GradSkip by handling more general regularizers and allowing general randomization. Last, it presents a method with variance reduction. The convergence of each method is analyzed. Numerical experiments illustrate the performance of GradSkip compared with the original method, ProxSkip.

**Audience:**

Yes

**Broader Impact Concerns:**

Nothing specific.

**Claims And Evidence:**

Yes

**Requested Changes:**

Nothing specific.

**Strengths And Weaknesses:**

The paper is well written. The ideas are clearly explained. There are both theoretical guarantees and numerical experiments.

---

> ### Author Response · Authors · 2025-01-08
> **Thanks**
>
> Thank you for your positive feedback.

---

### Review · Reviewer_peWp · 2025-01-07

**Summary Of Contributions:**

This paper develops new algorithms for distributed first-order optimization. When the different workers have different computational resources available, they use clever probabilistic and linear-algebraic ideas to optimize the loss function keeping low both the communication complexity and computational complexity.

They provide theoretical guarantees for their algorithms and show that their algorithms work well in practice as well, using both synthetic and real data.

**Audience:**

Yes

**Broader Impact Concerns:**

No concerns

**Claims And Evidence:**

Yes

**Requested Changes:**

Here are some minor suggestions to further improve the presentation.

1. In Definition 4.3, define the notation $\|\|x-y\|\|_L$.

2. In the caption of Figure 4, explain where the dataset *w6a* is coming from. Also, it's not clear why you have written 153 as a sum of factorials and as a sum of cubes. Wouldn't $n=153$ suffice?

**Strengths And Weaknesses:**

The paper is extremely clearly written, the motivation is explained well, intuition is given for the algorithms, and the experiments are comprehensive. The proposed algorithms are simple to implement, and the theoretical guarantees are strong.

---

> ### Author Response · Authors · 2025-01-09
> **Thanks**
>
> Thank you for your positive review and suggestions. We have addressed them in the revised version.

---

### Review · Reviewer_bc2o · 2025-03-19

**Summary Of Contributions:**

This proposes a redesign of the original ProxSkip method, allowing clients with ``less important'' data to get away with fewer local training steps without impacting the overall communication complexity of the method. The author prove that the modified method, GradSkip, converges linearly under the same assumptions and has the same accelerated communication complexity, while the number of local gradient steps can be reduced relative to a local condition number. The method is further generalized by extending the randomness of probabilistic alternations to arbitrary unbiased compression operators and by considering a generic proximable regularizer

**Audience:**

Yes

**Claims And Evidence:**

Yes

**Requested Changes:**

please refer to the weaknesses.

**Strengths And Weaknesses:**

Strengths:

1. Innovative Algorithm Design (GradSkip): The paper introduces GradSkip, an extension of the ProxSkip method by Mishchenko et al. (2022), which allows clients in federated learning (FL) to perform a variable number of local gradient steps based on their local data’s "importance" (condition number). This flexibility reduces computational complexity compared to ProxSkip, where all clients perform the same number of steps. tt maintains the accelerated communication complexity of ProxSkip (O(√κ_max log 1/ε)) while improving computational efficiency by allowing clients with well-conditioned problems (κ_i < √κ_max) to compute fewer local gradients (at most min(κ_i, √κ_max)).

2. Generalization (GradSkip+): The authors generalize GradSkip into GradSkip+, incorporating arbitrary proximable regularizers and unbiased compression operators with custom variance bounds. This makes the method versatile, recovering several existing methods (e.g., ProxGD, ProxSkip, RandProx-FB, and GradSkip itself) as special cases.

3. Variance Reduction (VR-GradSkip+): The paper proposes VR-GradSkip+, combining GradSkip with variance-reduced stochastic gradients (inspired by ProxSkip-VR). This reduces computational cost further by using mini-batch stochastic gradients instead of full-batch gradients, addressing both statistical heterogeneity and computational efficiency.

4. Theoretical Guarantees: The authors provide rigorous convergence proofs under strongly convex and smooth assumptions, showing linear convergence rates. For GradSkip, they demonstrate O(κ_max log 1/ε) iteration complexity and O(√κ_max log 1/ε) communication complexity, matching ProxSkip’s communication efficiency while improving computation.
Theorems (e.g., 3.6, 4.5, B.2) offer optimal parameter choices and quantify the computational savings, especially for heterogeneous systems.

Weaknesses:

1. Limited Scope of Convexity: A significant limitation, acknowledged in Section A, is that the theoretical guarantees apply only to strongly convex and smooth objectives. The communication acceleration property does not extend to non-strongly convex or nonconvex problems, limiting its applicability to many real-world machine learning tasks (e.g., deep learning).

2. Lack of Partial Participation: The paper assumes full client participation and does not address partial device participation, a common scenario in FL where only a subset of clients is active per round. Extending GradSkip to client sampling is noted as future work, indicating an incomplete solution for practical FL systems.

3. Theoretical Focus Over Empirical Validation: While the paper includes empirical studies, they are conducted on "carefully designed toy problems" rather than real-world datasets. This raises questions about how well GradSkip generalizes to complex, large-scale FL scenarios, a point not fully explored due to space constraints (e.g., VR-GradSkip+ details are in the appendix).

Complexity of Implementation: The algorithms (GradSkip, GradSkip+, VR-GradSkip+) introduce additional parameters (e.g., probabilities p, q_i, compression operators) and auxiliary variables (e.g., shifts h_i,t, ĥ_i,t), increasing implementation complexity compared to simpler methods like FedAvg. This could hinder adoption in resource-constrained environments.

4. Sensitivity to Parameter Tuning: Optimal performance relies on carefully chosen probabilities (e.g., q_i = (1 - 1/κ_i)/(1 - 1/κ_max), p = 1/√κ_max) and step sizes (e.g., γ = 1/L_max), which require knowledge of condition numbers (κ_i, κ_max). In practice, estimating these parameters accurately across heterogeneous clients may be challenging.

---

> ### Author Response · Authors · 2025-03-25
> **Addressing weaknesses 1**
>
> > Limited Scope of Convexity: A significant limitation, acknowledged in Section A, is that the theoretical guarantees apply only to strongly convex and smooth objectives. The communication acceleration property does not extend to non-strongly convex or nonconvex problems, limiting its applicability to many real-world machine learning tasks (e.g., deep learning).
>
>
> All theoretical work relies on assumptions—there is no free lunch. The assumptions we use are standard in the field and widely adopted in prior research. To the best of our knowledge, the phenomenon we study—accelerated communication with local training—holds under these assumptions, though its validity in more relaxed settings remains unknown. While this is a limitation, every theoretical paper has constraints. The key question is whether these limitations are reasonable for the problem at hand. In our case, they are. Using the same setup as ProxSkip, we refine the method and achieve better theoretical properties.
>
> Although neural network training is inherently non-convex, convex optimization remains essential for understanding non-convex problems and guiding heuristic improvements. Prior work [1] has shown that for certain neural networks, the regularized loss function is piecewise strongly convex on a key open set that, under some conditions, contains all global minimizers. This justifies our assumptions, which are both standard and widely used in theoretical research. Extending our analysis to non-convex settings remains an open challenge—even for ProxSkip—but the strongly convex and smooth regime is the only one where the benefits of local training have been rigorously established.
>
> Theory can be assessed along two dimensions: depth, which sharpens results within a specific setting, and breadth, which extends them to broader scenarios. Our focus is on depth, as meaningful generalizations require a solid foundation. While future work can explore extensions, our priority is to establish rigorous, insightful results that others can build upon.
>
>
> > Lack of Partial Participation: The paper assumes full client participation and does not address partial device participation, a common scenario in FL where only a subset of clients is active per round. Extending GradSkip to client sampling is noted as future work, indicating an incomplete solution for practical FL systems.
>
>
> Federated learning (FL) is typically categorized into two types [2]:
> - Cross-device FL, where clients are stateless because they are rarely seen more than once.
> - Cross-silo FL, where clients are stateful, as discussed in Sections 2.2 and 7.5 of [2].
>
> Our study focuses on cross-silo FL, as does most of the literature. In this setting, clients are fewer, more powerful, reliable, and known, which enables stateful methods. This distinction parallels the difference between single-pass and multiple-pass SGD: while single-pass methods see each data point only once, multiple-pass methods revisit data, making them potentially more powerful. Similarly, both cross-silo and cross-device FL are important and widely used in practice.
>
> A key aspect of our work is gaining a deeper understanding of local training (LT), a widely used but poorly understood technique in FL. FedAvg, the most common FL method, relies on LT along with data sampling and client sampling—yet these mechanisms, particularly LT, remain insufficiently understood. What does LT truly achieve, and how can it be improved? To answer this, we study LT in isolation, removing confounding factors like data sampling and partial client participation. This is necessary because even in this simplified setting, LT’s properties are not well understood.
>
> ProxSkip made progress in addressing this gap by formally establishing LT as a communication acceleration mechanism. It also demonstrated that accelerated LT can be combined with data sampling, though integrating it with partial client participation remained an open problem until recently. Rather than pursuing such extensions, we take a more fundamental approach: Can ProxSkip itself be improved? Specifically, can we maintain the same communication complexity while reducing the number of LT steps, adapting them to the importance of each client’s data? We provide a positive answer to this question.
>
> [1] Milne, Tristan. “Piecewise strong convexity of neural networks.” Advances in Neural Information Processing Systems 32 (2019).
> [2] Kairouz et al. Advances and Open Problems in Federated Learning, 2019. https://arxiv.org/pdf/1912.04977.pdf

---

> > ### Author Response · Authors · 2025-03-25
> > **Addressing weaknesses 2**
> >
> > > Theoretical Focus Over Empirical Validation: While the paper includes empirical studies, they are conducted on “carefully designed toy problems” rather than real-world datasets. This raises questions about how well GradSkip generalizes to complex, large-scale FL scenarios, a point not fully explored due to space constraints (e.g., VR-GradSkip+ details are in the appendix).
> >
> >
> > Our experiments are indeed simple by design. Their primary goal is to illustrate our theoretical findings rather than to address real-world deployment challenges. Since our work is theoretical, we believe that well-designed experiments provide clearer insights than large-scale empirical studies, which may introduce confounding factors.
> >
> > Moreover, theoretical contributions do not necessarily require large-scale experiments, just as practical works can be valuable without formal proofs. Our choice to focus on simple settings follows the principle of Occam’s razor—demonstrating key phenomena in the simplest possible way. While large-scale evaluations are important for applied research, they are beyond the scope of our study, which is not a systems-focused work.
> >
> >
> > > Complexity of Implementation: The algorithms (GradSkip, GradSkip+, VR-GradSkip+) introduce additional parameters (e.g., probabilities p, q_i, compression operators) and auxiliary variables (e.g., shifts h_i,t, ĥ_i,t), increasing implementation complexity compared to simpler methods like FedAvg. This could hinder adoption in resource-constrained environments.
> >
> > The ProxSkip paper demonstrated that shifts are essential for achieving accelerated convergence guarantees. This is particularly important in heterogeneous settings, where devices have different local data distributions. In such cases, local training introduces local drift, and shifts are necessary to correct this drift.
> >
> > The only additional parameter in GradSkip compared to ProxSkip is the probability $q_i$, which is simply a real number in $[0,1]$. This does not increase implementation complexity or impose additional resource constraints.
> >
> > > Sensitivity to Parameter Tuning: Optimal performance relies on carefully chosen probabilities (e.g., q_i = (1 - 1/κ_i)/(1 - 1/κ_max), p = 1/√κ_max) and step sizes (e.g., γ = 1/L_max), which require knowledge of condition numbers (κ_i, κ_max). In practice, estimating these parameters accurately across heterogeneous clients may be challenging.
> >
> >
> > In GradSkip, we use the local smoothness constants $\kappa_i$ to account for the relative importance of local data. However, knowing these constants is not a strict requirement—this is the same assumption that standard gradient descent makes when optimizing local functions. Therefore, this requirement is not particularly restrictive. While adaptivity is an interesting direction, it is beyond the scope of our work.
> >
> > Furthermore, in the experiment shown in Figure 4, we do not use the theoretically derived probabilities that depend on $\kappa_i$. Instead, we select them based on worker speed and observe that our method remains robust to these choices. This suggests that GradSkip performs well even when the exact values of $\kappa_i$ are unknown.

---

### Decision · Action_Editor_RqYb · 2025-06-04

**Recommendation:** Accept as is

**Audience:**

Yes

**Audience Explanation:**

The paper proposes new communication-efficient algorithms for federated learning, which is of broad relevance to TMLR's audience.

**Claims And Evidence:**

Yes

**Claims Explanation:**

The reviewers unanimously commend the paper for its clarity, the relevance of its contributions, and the strong support for its claims through both rigorous theoretical analysis and comprehensive numerical experiments.